# Neural Program Generation Modulo Static Analysis

**Rohan Mukherjee**
Rice University

**Yeming Wen**
UT Austin

**Dipak Chaudhari**
UT Austin

**Thomas W. Reps**
University of Wisconsin

**Swarat Chaudhuri**
UT Austin

**Chris Jermaine**
Rice University

## Abstract

State-of-the-art neural models of source code tend to be evaluated on the generation of individual expressions and lines of code, and commonly fail on long-horizon tasks such as the generation of entire method bodies. We propose to address this deficiency using weak supervision from a static program analyzer. Our neurosymbolic method allows a deep generative model to symbolically compute, using calls to a static-analysis tool, long-distance semantic relationships in the code that it has already generated. During training, the model observes these relationships and learns to generate programs conditioned on them. We apply our approach to the problem of generating entire Java methods given the remainder of the class that contains the method. Our experiments show that the approach substantially outperforms state-of-the-art transformers and a model that explicitly tries to learn program semantics on this task, both in terms of producing programs free of basic semantic errors and in terms of syntactically matching the ground truth.

## 1 Introduction

Neural models of source code have received much attention in the recent past [38, 9, 26, 23, 30, 36, 16, 24]. However, these models have a basic weakness: while they frequently excel at generating individual expressions or lines of code, they do not do so well when tasked with synthesizing larger code blocks. For example, as we show later in this paper, state-of-the-art transformer models [8, 6, 24] can generate code with elementary semantic errors, such as uninitialized variables and type-incorrect expressions, when asked to generate *method bodies*, as opposed to single lines. Even in terms of syntactic accuracy measures, the quality of the code that transformers produce on such "long-horizon" tasks can be far removed from the ground truth.

The root cause of these issues, we believe, is that current neural models of code treat programs as text rather than artifacts that are constructed following a *semantics*. In principle, a model could learn semantics from syntax given enough data. In practice, such learning is difficult for complex, general-purpose languages.

In this paper, we propose to address this challenge through an alternative *neurosymbolic* approach. Our main observation is that symbolic methods—specifically, static program analysis—can extract deep semantic relationships between far-removed parts of a program. However, these relationships are not apparent at the level of syntax, and it is difficult for even large neural networks to learn them automatically. Driven by this observation, we use a static-analysis tool as a *weak supervisor* for a deep model of code. During generation, our model invokes this static analyzer to compute a set of semantic facts about the code generated so far. The distribution over the model's next generation actions is conditioned on these facts.

We concretely develop our approach by extending the classic formalism of *attribute grammars* [20]. Attribute grammars are like context-free grammars but allow rules to carry symbolic *attributes* of the

35th Conference on Neural Information Processing Systems (NeurIPS 2021).

context in which a rule is fired. In our model, called *Neurosymbolic Attribute Grammars* (NSGs), the context is an incomplete program, and rules are fired to replace a nonterminal (a stand-in for unknown code) in this program. The attributes are semantic relationships (for example, symbol tables) computed using static analysis. The neural part of the model represents a probability distribution over the rules of the grammar conditioned on the attributes. During generation, the model repeatedly samples from this distribution while simultaneously computing the attributes of the generated code.

We evaluate our approach in the task of generating the entire body of a Java method given the rest of the class in which the method occurs. We consider a large corpus of curated Java programs, over a large vocabulary of API methods and types.Using this corpus, we train an NSG whose attributes, among other things, track the state of the symbol table and the types of arguments and return values of invoked methods at various points of a program, and whose neural component is a basic tree LSTM. We compare this model against several recent models: fine-tuned versions of two GPT-NEO [6] transformers and the CODEGPT [24] transformer, OpenAI's CODEX system [8] (used in a zero-shot manner), and a GNN-based method for program encoding [7]. Some of these models are multiple orders of magnitude larger than our NSG model. Our experiments show that the NSG model reliably outperforms all of the baselines on our task, both in terms of producing programs free of semantic errors and in terms of matching the ground truth syntactically.

In summary, this paper makes three contributions:

- We present a new approach to the generative modeling of source code that uses a static-analysis tool as a weak supervisor.

- We embody this approach in the specific form of *neurosymbolic attribute grammars* (NSGs).

- We evaluate the NSG approach on the long-horizon task of generating entire Java method bodies, and show that it significantly outperforms several larger, state-of-the-art transformer models.

## 2   Conditional Program Generation

We start by stating our problem, known as conditional program generation (CPG) [26]. We imagine a joint distribution $\mathcal{D}(X, Y)$, where $X$ ranges over *specifications* of program-generation problems and $Y$ ranges over programs. The probability $\mathcal{D}(X = \mathsf{X}, Y = \mathsf{Y})$ is high when $\mathsf{Y}$ is a solution to $\mathsf{X}$. Also, we consider a family of distributions $\mathcal{P}_\theta(Y|X = \mathsf{X})$, parameterized by $\theta$, that we might want to learn. *Learning to conditionally generate* programs amounts to finding parameters $\theta$ that minimize the prediction error $\mathbf{E}_{(\mathsf{X},\mathsf{Y}) \sim \mathcal{D}}[\delta(\mathcal{P}_\theta(\mathsf{X}|\mathsf{Y}), \mathsf{Y})]$, where $\delta$ is a suitable distance function between programs.

Specifications and distances between programs can be defined in many ways. In our experiments, the goal is to generate Java method bodies. A specification is an *evidence set* that contains information—e.g., method names, types of variables and methods—about the class in which the method lies. We define $\delta(\mathsf{Y}_1, \mathsf{Y}_2)$ to be a large number if $\mathsf{Y}_1$ or $\mathsf{Y}_2$ violates one of several language-level invariants (e.g., type-safety, initialization of variables before use) that we require programs to satisfy. When both programs satisfy the invariants, $\delta(\mathsf{Y}_1, \mathsf{Y}_2)$ measures the textual dissimilarity between the two programs.

```
(a)

public class FileUtil{
  String err;
  public int read(File f){...}

  /* write lines to file */
  public void write(
    File f, String str){??}}

(b)

void write(File f, String str){
  try {
    FileWriter var_0;
    var_0 = new FileWriter(f);
    var_0.write(str);
  } catch(IOException var_0) {
    var_0.printStackTrace();
    System.out.println( ARG ); }
  return; }
```

Figure 1: (a) An instance of conditional program generation. (b) A top completion of the `write` method, generated using an NSG. `ARG` stands for a string literal.

Note that CPG is a much more challenging task than the well-studied next-token-prediction task [24, 7]. The goal is to predict long sequences of tokens (e.g., an entire method body). Also, $\mathsf{X}$ is a (possibly imprecise) specification of the code to generate, not just a sequence of tokens we are trying to complete by, say, choosing the correct method to call for a variable.

**Example.** Fig. 1-(a) illustrates the kind of task that we target. Here, we are given a class with a missing `write` method. The specification $\mathsf{X}$ includes: (i) the class name `FileUtil`; (ii) the

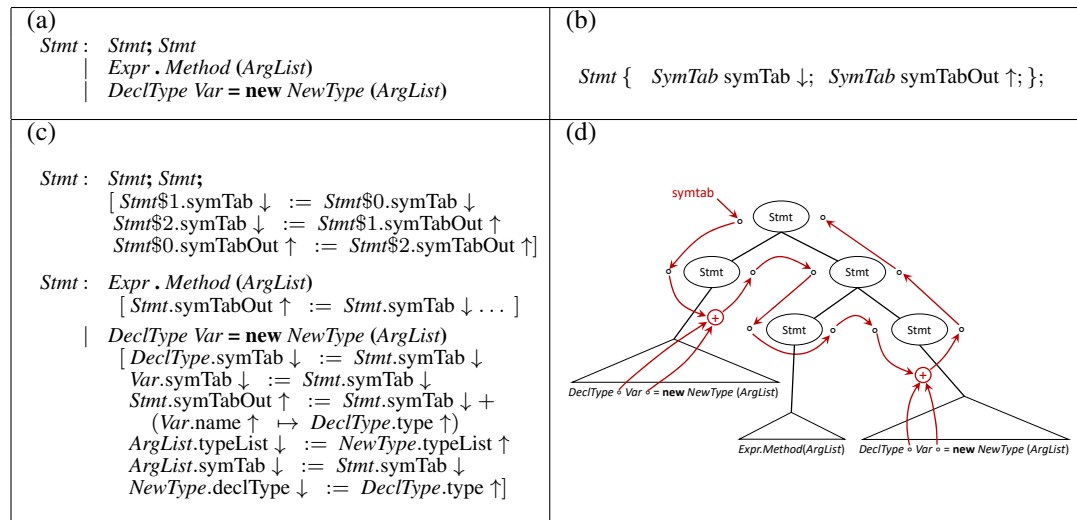

Figure 2: (a) A basic context-free grammar. (b) Attributes of the *Stmt* nonterminal. (c) Attribute equations for the productions (the parts of the equations denoted by "..." are elided). (d) An attributed tree, illustrating left-to-right threading of attributes.

type `String` of the class variable `err`; (iii) information about complete methods within the class (including the methods' return types and formal-parameter types and names, and sequences of API calls made within such methods); (iv) information about the method with missing code (`write`), including its name, formal parameters, and JavaDoc comments for the method with missing code (e.g., "write lines to file"). Our objective on this input is to generate automatically a non-buggy, natural completion of `write`, without any provided, partial implementation of the method.

To understand the challenges in this task, consider a completion that starts by: (i) declaring a local variable `var_0`; and (ii) invoking the constructor for `FileWriter` and storing the result in `var_0`. A proper implementation of these two steps must ensure that `var_0` is of type `FileWriter`. Also, the first argument to the constructor of `FileWriter` must be of type `File` (or a subtype of `File`). As we show in Sec. 5, it is hard for state-of-the-art neural models to learn to satisfy these rules.

In contrast, in our approach, the generator has access to a set of semantic attributes computed via static analysis. These attributes include a *symbol table* mapping in-scope variables to their types.

Suppose that during training we are given the following line of code: "`var_0 = new FileWriter(f, true)`". Our model's symbol table includes the names `var_0` and `f` and their types. The grammar is also able to compute the type of the first argument in the invoked constructor for `FileWriter`. Consequently, the model can observe that the type of `f` is listed as `File` in the symbol table, and that `f` is the first argument to the `FileWriter` constructor. With a few observations like these, the model can learn that the first argument of "`new FileWriter`" tends to be of type `File` (or a subtype). During generation, the model uses this knowledge, locating a variable of the correct type in the symbol table each time it constructs a `FileWriter`.

Fig. 1-(b) shows a top completion of `write` generated by our NSG implementation. Note that all variables in this code are initialized before use, and that all operations are type-safe. Also, the name `var_0` is reused between the **try** and the **catch** blocks. Such reuse is possible because the symbol table carries information about the scopes to which different names belong. Finally, as we will see in Sec. 5, the extra information provided by the static analyzer can also help with accuracy in terms of syntactic matches with the ground truth.

# 3 Static Analysis with Attribute Grammars

As mentioned in Sec. 1, we develop our approach as an extension of the classic attribute grammar (AG) framework [20]. Now we give some background on static analysis using AGs. In the next section, we show how to use AGs to weakly supervise a neural program generator.

An AG extends a traditional context-free grammar (CFG) [18] by attaching a set of *attributes* to each terminal or nonterminal symbol of the grammar and by using a set of *attribute equations* to propagate attribute values through syntax trees. The attributes of a symbol $S$ can be divided into *inherited* attributes and *synthesized* attributes, which we suffix by $\downarrow$ and $\uparrow$, respectively. Inherited attributes transfer information from parent to child, or from a node to itself. Synthesized attributes transfer information from child to parent, from a node to a sibling, or from a node to itself. We assume that the terminal symbols of the grammar have no synthesized attributes and that the root symbol of the grammar has a special set of inherited attributes, known as the *initial attributes*.

The *output* attributes of a production $S \to S_1, \ldots, S_k$ consist of the synthesized-attribute occurrences of the nonterminal $S$, plus the inherited-attribute occurrences of all of the $S_i$'s. The *input* attributes are the inherited-attribute occurrences of $S$, plus the synthesized-attribute occurrences of the $S_i$'s. The grammar's attribute equations relate the input and output attributes of a node in terms of the attributes of its parent, children, and left sibling in the syntax tree that the grammar generates.

**Example.** Consider the simple CFG in Fig. 2-(a). The nonterminal *Stmt* stands for *program statements*. The grammar says that a statement can either be a sequential composition of statements, a method call, or a variable declaration. A natural AG extension of this CFG tracks symbol tables, which allow easy lookup of all variables in scope.

Specifically, the grammar associates two symbol-table-valued attributes, symTab $\downarrow$ and symTabOut $\uparrow$, with *Stmt* (Fig. 2-(b)). The attributes are propagated following the equations in Fig. 2-(c). In these equations, we distinguish between the three different occurrences of nonterminal "*Stmt*" via the symbols "*Stmt*\$0," "*Stmt*\$1," and "*Stmt*\$2." where the numbers denote the leftmost occurrence, the next-to-leftmost occurrence, etc. In this case, the leftmost occurrence is the left-hand-side occurrence.

For concreteness, let us consider the attribute equations for the production for sequential composition in the grammar. Here, the inherited attribute of *Stmt*\$0 gets passed "down" the syntax tree as an inherited attribute of *Stmt*\$1. The synthesized attribute received at *Stmt*\$1 is passed to *Stmt*\$2 as an inherited attribute. More generally, the attribute equations define a left-to-right information flow through the syntax tree, as illustrated in Fig. 2-(d).

## 4 Neurosymbolic Attribute Grammars

Now we introduce the model of neurosymbolic attribute grammars (NSGs). Our goal is to learn a distribution $\mathcal{P}(Y|X)$, where $Y$ is a random variable whose domain is all possible programs (concretely, Java method bodies) and $X$ is a specification of a program-generation problem (concretely, an *evidence set* made up of useful information extracted symbolically from the method's context and then encoded using a neural network). Attributes containing the results of a symbolic, static analysis are available to the neural network implementing this distribution. This weak supervision allows the network to mimic more accurately the long-range dependencies present in real code-bases.

**The Underlying Model.** The idea of weak supervision using a static analyzer could be developed on top of many different kinds of models. Here, we develop the idea on top of a model from Murali et al. [26]. This model uses a latent variable $Z$ to represent the *true user intent* behind the incomplete or ambiguous evidence set $Y$. We then have $\mathcal{P}(Y|X) = \int_Z \mathcal{P}(Z|X)\mathcal{P}(Y|Z)dZ$. To define the distribution $\mathcal{P}(Z|X)$, we assume that the evidence set has data of a fixed number of *types*—e.g., method names, formal parameters, and Javadoc comments.

The $j^{th}$ type of evidence has a neural encoder $f_j$. An individual piece of evidence $X$ is either encoded as a single vector or as a set of vectors with no particular ordering. For example, our implementation encodes Javadoc comments as vectors using LSTMs, and each member of a set of formal parameters using a basic feedforward network. Let $X_{j,k}$ refer to the $k^{th}$ instance of the $j^{th}$ kind of evidence in $X$. Assume a Normal prior on $Z$, and let $\mathcal{P}(X|Z) = \prod_{j,k} \mathcal{N}\left(f_j(X_{j,k}) \mid Z, \mathbf{I}\sigma_j^2\right)$. Assume that the encoding of each type of evidence is sampled from a Normal centered at $Z$. If $f$ is 1-1 and onto, we have [26]:

$$\mathcal{P}(Z|X) = \mathcal{N}\left(Z \mid \frac{\sum\limits_{j,k} \sigma_j^{-2} f_j(X_{j,k})}{1 + \sum\limits_{j} |X_j|\sigma_j^{-2}}, \frac{1}{1 + \sum\limits_{j} |X_j|\sigma_j^{-2}}\mathbf{I}\right)$$

Next, we define the distribution $\mathcal{P}(Y|Z)$. Consider a stochastic CFG which assumes (1) that a leftmost derivation is carried out, and (2) the probability distribution governing the expansion of a symbol in the grammar takes into account the sequence of all expansions so far, as well as an input value $Z$ upon which all expansions are conditioned.

This CFG consists of productions of the form $S : seq_1 | seq_2 | seq_3... | seq_n$. Each symbol such as $S$ corresponds to a categorical random variable with sample space $\Omega(S) = \{seq_1, seq_2, ..., seq_n\}$. A trial over the symbol $S$ randomly selects one of the RHS sequences for that symbol. If $S$ is a terminal symbol, then $\Omega(S) = \{\epsilon\}$, where $\epsilon$ is a special value that cannot be expanded. Subsequently, when a trial over $S$ is performed and an RHS sequence from $\Omega(S)$ is randomly selected, we will use the sans-serif $\mathsf{S}^{\text{rhs}}$ to denote the identity of the RHS sequence observed.

Now consider a depth-first, left-to-right algorithm for non-deterministically expanding rules in the grammar to generate a *program* $Y = \langle (S_1, \mathsf{S}_1^{\text{rhs}}), (S_2, \mathsf{S}_2^{\text{rhs}}), ... \rangle$; here, each $S_i$ is a symbol encountered during the expansion, and each $\mathsf{S}_i^{\text{rhs}}$ is the identity of the RHS chosen for that symbol. Let $S_1$ correspond to the symbol Start. We perform a trial over $S_1$ and

---

**Algorithm 1:** $\text{Gen}(S, A(S)\downarrow, \text{SymSoFar}, Z)$

---

**Input**: current symbol $S$, inherited attributes $A(S)\downarrow$, sequence of symbols so far SymSoFar, latent encoding Z

**Modifies**: all symbols expanded are appended to SymSoFar

**Returns**: $A(S)\uparrow$, the synthesized attrs of $S$

**if** $S$ *is a terminal symbol* **then**
    Append $(S, \epsilon)$ to SymSoFar
    **return** $\emptyset$
**else**
    Choose a right-hand-side (RHS) sequence $\mathsf{S}^{\text{rhs}} \sim \mathcal{P}(S|\text{SymSoFar}, A(S)\downarrow, Z)$
    Append $(S, \mathsf{S}^{\text{rhs}})$ to SymSoFar
    SynthSoFar $\leftarrow \langle \rangle$
    **for** $S' \in \mathsf{S}^{rhs}$ *in left-to-right order* **do**
        Compute $A(S')\downarrow$ from $A(S)\downarrow$ and SynthSoFar
        $A(S')\uparrow \leftarrow$ $\text{Gen}(S', A(S')\downarrow, \text{SymSoFar}, Z)$
        Append $A(S')\uparrow$ to SynthSoFar
    **end**
**end**
Compute $A(S)\uparrow$ from $A(S)\downarrow$ and SynthSoFar
**return** $A(S)\uparrow$

---

select one of the RHS sequences from $\Omega(S_1)$. Let the identity of the RHS sequence selected be $\mathsf{S}_1^{\text{rhs}}$. Note that $\mathsf{S}_1^{\text{rhs}}$ is *itself* a sequence of symbols. Choose the first symbol in the sequence $\mathsf{S}_1^{\text{rhs}}$; call this symbol $S_2$. Perform a trial over $S_2$, and let the identity of the RHS sequence chosen be $\mathsf{S}_2^{\text{rhs}}$. Choose the first symbol in $\mathsf{S}_2^{\text{rhs}}$ (call it $S_3$) and expand it the same way. This recursive descent continues until a terminal symbol $S_i$ is encountered, and the recursion unwinds. If the recursion unwinds to symbol $S_2$, for example, then we choose the second symbol in the sequence $\mathsf{S}_1^{\text{rhs}}$, which we call $S_{i+1}$. We perform a trial over $S_{i+1}$, and let the identity of the RHS sequence chosen be $\mathsf{S}_{i+1}^{\text{rhs}}$. This sequence is recursively expanded. Once all of the symbols in the RHS associated with the Start symbol $S_1$ have been fully expanded, we have a program.

This generative process defines a probability distribution $\mathcal{P}(Y|Z)$, where for a particular program $Y$, the probability of observing $Y$ is computed as

$$\mathcal{P}(Y|Z) = \prod_i \mathcal{P}(S_i = \mathsf{S}_i^{\text{rhs}}|S_1 = \mathsf{S}_1^{\text{rhs}}, ..., S_{i-1} = \mathsf{S}_{i-1}^{\text{rhs}}, Z). \tag{1}$$

We henceforth abbreviate the expression for the inner probability as $\mathcal{P}(\mathsf{S}_i^{\text{rhs}}|\mathsf{S}_1^{\text{rhs}}, ..., \mathsf{S}_{i-1}^{\text{rhs}}, Z)$.

**Weak Supervision with Attributes.** Now assume that the grammar is an AG, so that each symbol $S$ has an attribute set $A(S)$. We use $A(S)\uparrow$ to denote the synthesized attributes of $S$, and $A(S)\downarrow$ to denote the inherited attributes of $S$.

An NSG extends the model so that the conditional distribution $\mathcal{P}(Y|Z)$ is defined as:

$$\mathcal{P}(Y|Z) = \prod_i \mathcal{P}(\mathsf{S}_i^{\text{rhs}}|\langle \mathsf{S}_1^{\text{rhs}}, \mathsf{S}_2^{\text{rhs}}, ..., \mathsf{S}_{i-1}^{\text{rhs}}\rangle, A(S_i)\downarrow, Z).$$

That is, when a symbol $S_i$ is non-deterministically expanded, its value depends not just on the latent position $Z$ and the sequence of expansions thus far, but also on the values of $S_i$'s inherited attributes, $A(S_i)\downarrow$. In theory, a powerful enough learner with enough data could learn the importance of these sets of attribute values, without ever seeing them explicitly. In that sense, they could be treated as latent variables to be learned. However, the benefit of having a static analysis produce these values

deterministically is that the author of a static analysis knows the semantic rules that must be followed by a program; by presenting the data used to check whether those rules are followed directly to a learner, the process of learning to generate programs is made much easier.

Generation of a program under an NSG is described in Algorithm 1, where the distribution governing the expansion of symbol $S$ has access to attribute values $A(S)\downarrow$.

**Designing an appropriate static analysis.** Intuitively, a program generated with the supervision of a static analyzer is likely to generate a semantically correct program because the static analysis provides key semantic clues during program generation. In a conventional AG-based analyzer, the AG would be used to maintain data structures that can be used to *validate* that in a complete program, key relationships hold among the values of the production's attributes. Our goal is to generate programs, rather than validate them; also, we want to *guide* the learner rather than impose hard constraints. However, constraints are a good mental model for designing a good NSG. That is, we generally expect the attribute equations used at important decision points during a neural generation process to be also helpful for validating key semantic properties of complete programs.

**Example.** Now we show how to use the attribute grammar in Fig. 2 in generating the body of the `write` method from Sec. 2. Let us assume that the grammar has a start nonterminal *Start* (not shown in Fig. 2) that appears in a single rule expanding it to the statement nonterminal *Stmt*. We start by extracting the context X around the method, then use this information to sample Z from $P(Z|\mathsf{X})$. Next, a Java compiler processes the surrounding code and the method's formal parameters to form the attributes $A(Start)\downarrow$, which we assume to consist of a symbol table { f $\mapsto$ File, str $\mapsto$ String }.

To generate a program, we sample from the distribution $P(Start|\langle\rangle, A(Start)\downarrow, \mathsf{Z})$. First, *Start* is expanded to "*Stmt* **;** *Stmt*". When expanding the first *Stmt*, the NSG needs to choose between a method invocation and a variable declaration. Because the NSG is "aware" that this step is to expand the first line of the method—the list of RHS values chosen so far is empty—we would expect it to declare a variable. This choice gives us the RHS "*DeclType Var* **=** **new** *NewType* (*ArgList*)". Expanding *DeclType*, the NSG samples a Java type from the distribution

$$P(DeclType|\langle\text{"}Stmt\text{; }Stmt\text{"}, \text{"}DeclType\ Var = \textbf{new}\ NewType\ (ArgList)\text{"}\rangle, A(DeclType)\downarrow, \mathsf{Z}).$$

From the rules for expanding the nonterminal *DeclType* in Fig. 2, we see that the NSG can choose any Java type as the declared type of the variable. At this point, the NSG is aware that the goal is to create a method called `write` (this is encoded in Z) and that it is choosing a type to be declared on the first line of the method. It also has access to the symbol table that is maintained as part of $A(DeclType)\downarrow$. Thus, the NSG may decide to expand the symbol *DeclType* to `FileWriter`. This type is then passed upward via the synthesized attribute *DeclType*.type$\uparrow$.

Next, the grammar must expand the *Var* rule and pick a variable name to declare. This choice is returned via the synthesized attribute *Var*.name$\uparrow$. Now it is time to expand *NewType*. The attributes make this easy: when sampling from $P(NewType|...)$, the NSG has access to *NewType*.type$\downarrow$, which takes the value `FileWriter`. A synthesizer may err by choosing a type that is not compatible with `FileWriter`. However, we may expect that during training, every time that *NewType* was expanded and the declared type was `FileWriter`, the type chosen was either `FileWriter` or some subclass of `FileWriter`. Hence the NSG is unlikely to make an error.

Assume that the NSG chooses `FileWriter`. It must now expand *ArgList*. Again, the NSG has the advantage of having access to *ArgList*.typeList$\downarrow$ (an explicit representation of the types required by the constructor being called) and, most importantly, *ArgList*.symTab$\downarrow$ (an explicit list of the variables in scope, as well as their types). At this point, it is easy for the NSG to match the required type of the first argument to the constructor (`File`) with an appropriate variable in the symbol table (`f`).

Now that the declaration of `var_0` has been fully expanded, the NSG updates the symbol table with a binding for the newly-declared variable `var_0`, and the attribute *Stmt*.symTab$\uparrow$ takes the value $\{$f $\mapsto$ File, str $\mapsto$ String, var_0 $\mapsto$ FileWriter$\}$. When the second occurrence of *Stmt* is expanded, the symbol table is passed down via the inherited attribute *Stmt*$1.symTab \downarrow$. All of the information available—the latent variable Z encoding the contextual information (including the name of the method "`write`" being generated), and the symbol table containing a `FileWriter` and a `String`)—helps the NSG to deduce correctly that this *Stmt* symbol should be expanded into an invocation of a `write` method. Also, the presence of the symbol table makes it easy for the NSG to correctly attach the `write` method call to the variable `var_0` and to use `str` as the argument.

# 5 Evaluation

Our experimental hypothesis is that neural networks find it difficult to learn the intricate rules that govern the generation of code by only looking at the syntax of example programs. These issues become especially visible when the units of code to be generated are large, for example, entire method bodies. In contrast, an NSG can use its static analysis to compute long-distance dependencies between program variables and statements "for free." Because of this extra power, NSGs can outperform much larger neural models at generating accurate and semantically correct code.

## 5.1 Experimental Setup

**Data**. To test our hypothesis, we used a curated, deduplicated set of Java source-code files [26]. For each class and each method, we used the remainder of the class as evidence or context, and the method body was used to produce training or test data. We used 1.57 M method bodies for training. The grammar used had ten terminals corresponding to formal parameters, ten for class variables, and ten for methods local to the class. None of the Java classes in the corpus needed more than ten of each of these terminals; when generating training data, each declared Java variable or method was randomly mapped to one of the appropriate terminals. Approximately 8,000 types and 27,000 method calls from the Java JDK also appeared as terminals in the grammar.

**NSG Implementation.** We implemented an NSG for our subset of Java. Here, attributes are used to keep track of the state of the symbol table, the expected return type of each method, expected types of actual parameters, variable initialization, whether the variable has been used, and whether the method has a return statement. The symbol table contains entries for all formal parameters, class variables, and internal methods within the class.

The neural part of our model has 63 M parameters. To expose the attributes to the neural part of the model, we implement a depth-first search over a program's abstract syntax tree (AST) to extract node information. The attributes are then encoded in a standard way — for example, the symbol table is represented as matrix (rows correspond to types, columns to variables, the value 1 is present if the corresponding type/variable pair is in scope). The distribution $\mathcal{P}(\mathsf{S}_i^{\mathrm{rhs}}|\langle\mathsf{S}_1^{\mathrm{rhs}}, \mathsf{S}_2^{\mathrm{rhs}}, ..., \mathsf{S}_{i-1}^{\mathrm{rhs}}\rangle, A(S_i)\downarrow, \mathsf{Z})$ is implemented as a set of LSTMs that decode the sequence of symbols, as well as the encoded $A(S_i)\downarrow$ and $\mathsf{Z}$, into a distribution over $\mathsf{S}_i^{\mathrm{rhs}}$. We trained our framework on top of Tensorflow [1]. Using one GPU, the NSG training time is around 72 hours. See Appendix C for more details.[1]

**Baselines.** We consider three categories of baselines. The first consists of large pretrained transformers. Specifically, we consider two variants of the GPT-NEO [6] model with 125 M and 1.3 B parameters. Both models are pre-trained on the Pile dataset [15], which consists of an 800 GB English-text corpus and open-source code repositories. On the APPS dataset [17], they perform well compared to OpenAI's 12-B-parameter, GPT-3-like CODEX model [8]. We also compare against CODEGPT [24] which is a GPT-2-like model with 125 million parameters. This model was pre-trained on Python and Java corpora from the CodeSearchNet dataset, which consists of 1.1 M Python functions and 1.6 M Java methods. We fine-tune all of these pretrained models on our Java dataset, using the token-level code-completion task provided by CodeXGLUE [24]. Finally, we also offer a comparison against the CODEX model [8]. Because we did not have access to the model's pretrained weights, this model is only used in a zero-shot fashion (no fine-tuning on our Java dataset). It should be noted here that the transformer baselines work on the entire Java language, whereas our NSG framework works on a sub-part of Java which is supported in our grammar definition.

The second category comprises an ablation, called a "conditional neural grammar" (CNG), that is identical to our NSG model but is trained without any of the attribute information. In other words, the CNG model is trained only on the program syntax. The third category includes GNN2NAG [7], a graph-neural-network-based method that uses an attribute grammar but *learns* the attributes from data rather than computing them symbolically. See Appendix C for more details on the baselines.

**Test Scenario.** Our test scenario is as follows. Given a Java class, we remove the entire body of a randomly selected method. We then use the remaining potion of the class along with the method header as context information that is then fed to the model as input. We run our NSG model and the baselines to regenerate this method body conditioned on the resulting context. We report the accuracy of the prediction based on static-semantic checks and fidelity measures.

---

[1]Our implementation is available at `https://github.com/rohanmukh/nsg`.

Table 1: Percent of Static Checks Passed

| | GPTNeo125M | GPTNeo1.3B | CODEX | CODEGPT | GNN2NAG | CNG | NSG |
|---|---|---|---|---|---|---|---|
| No undeclared variable access | 89.87% | 90.36% | 88.62% | 90.94% | 47.44% | 19.78% | **99.82%** |
| Valid formal parameter access | NA | NA | NA | NA | 25.78% | 11.03% | **99.55%** |
| Valid class variable access | NA | NA | NA | NA | 15.40% | 12.75% | **99.53%** |
| No uninitialized objects | 93.90% | 91.73% | 90.82% | 94.37% | 21.20% | 21.56% | **99.01%** |
| No variable access error | 90.36% | 90.51% | 88.86% | 91.32% | 28.92% | 17.92% | **99.69%** |
| Object-method compatibility | **98.36%** | 98.09% | 98.35% | 97.84% | 21.43% | 12.23% | 97.53% |
| Return type at call site | 97.38% | 98.01% | **98.53%** | 97.83% | 23.86% | 16.40% | 98.01% |
| Actual parameter type | 87.03% | 86.36% | 92.28% | 88.71% | 9.27% | 16.09% | **97.96%** |
| Return statement type | 84.05% | 85.09% | 88.13% | 85.23% | 12.34% | 9.51% | **90.97%** |
| No type errors | 87.25% | 88.13% | 91.42% | 88.10% | 16.31% | 13.56% | **97.08%** |
| Return statement exists | 99.61% | 99.80% | 98.44% | 99.57% | 94.02% | **99.92%** | 97.10% |
| No unused variables | 96.42% | 96.46% | 96.82% | **97.64%** | 20.95% | 24.29% | 93.84% |
| Percentage of parsing | 98.18% | 98.13% | 96.41% | 97.08% | **100.0%** | **100.0%** | **100.0%** |
| Pass all checks | 65.26% | 64.88% | 47.49% | 67.73% | 17.34% | 12.87% | **86.41%** |

Table 2: Average Fidelity of Generated Method Bodies

| | GPTNeo125M | GPTNeo1.3B | CODEX | CODEGPT | GNN2NAG | CNG | NSG |
|---|---|---|---|---|---|---|---|
| Set of API Calls | 32% | 37% | 36% | 36% | 3% | 22% | **53%** |
| Sequences of API Calls | 17% | 20% | 16% | 19% | 0.3% | 18% | **42%** |
| Sequences of Program Paths | 12% | 15% | 10% | 14% | 0% | 17% | **39%** |
| AST Exact Match | 12% | 15% | 10% | 14% | 0% | 6% | **26%** |

## 5.2 Results

**Static Checks.** For each generated method body, we check the following properties: *(1) No undeclared variable access:* Are all the variables used in a program declared (within an enclosing scope) before they are used? *(2) Valid formal parameter access:* Are formal parameters that are used in the method body present in the method declaration? *(3) Valid class-variable access:* Are the class variables that are used in the method body present in the class declaration? *(4) No uninitialized objects:* Do variables have a non-null value when they are used? *(5) No variable access errors:* Are checks (1)-(4) all satisfied? *(6) Object-method compatibility:* Are methods called on objects of a given class actually available within that class? *(7) Return type at the call site:* Is the assignment of the return value type-correct with respect to the return type of the called method? *(8) Actual-parameter type:* Are the actual-parameter types in an API call consistent with the corresponding formal-parameter types? *(9) Return-statement type:* Is the type of the expression in a return statement consistent with the method's declared return type? *(10) No type errors:* Are checks (6)-(10) all satisfied? *(11) Return statement exists:* Does the method body have a return statement somewhere? *(12) No unused variables:* Are all variables declared in the method body used in the method? *(13) Percentage of parsing:* Can the generated method be parsed by a standard Java parser? *(14) Pass all checks:* Are checks (1)-(13) all satisfied?

Note that (2) and (3) are not meaningful metrics for approaches, such as our transformer baselines, that do not use a grammar to generate code. This is because in these models, when a variable token is generated, there is no way to tell what category of variable (class variable, formal parameter, etc.) it is meant to be. These metrics are meaningful for the NSG, CNG, and GNN2NAG models, which use a Java parser capable of partitioning variable names into different categories.

The results of our comparisons appear in Table 1. These scores are interpreted as follows. Suppose that a generated program uses five variables, of which four are declared correctly in the proper scope. This situation is scored as 80% correct on the "No undeclared-variable access" criterion. We report the average success rate over each of these properties over all the generated programs in our test suite.

**Whole-Method Fidelity.** We also check the *fidelity* of the generated code to the reference code. One possibility here is to use a standard metric for text generation, such as the BLEU score. However, this is problematic. As the BLEU score is not invariant to variable renamings, a nonsensical program that uses commonplace variable names can get an artificially high BLEU score. Also, programs are structured objects in which some tokens indicate control flow and some indicate data flow. The BLEU score does not take this structure into account. See Appendix J for a concrete example of these issues.

| Query | ```java
public class FileUtils{
  FileReader field_7;
  BufferedReader field_5;

  /** read line from file */
  public String reader () {}}
``` |
|---|---|
| NSG | ```java
public String reader(){
  java.lang.String var_9;
  try {var_9=field_5.readLine();
  } catch (IOException var_8) {
   var_8.printStackTrace(); }
  return var_9; }
``` |
| CodeGPT | ```java
public String reader(){
    StringBuffer buffer=
      new StringBuffer();
    buffer.append("\n");
    return buffer.toString();
}
``` |
| CODEX | ```java
public String reader()
 throws IOException{
  field_5= new BufferedReader(
   new FileReader(field_7));
  return field_5.readLine(); }
``` |
| GptNeo1.3B | ```java
public String reader()
 throws IOException {
  try {
   field_7=new FileReader(this.file);
   field_5=new BufferedReader(field_7);
   String line;
   while (field_5.readLine()) {
    System.out.println(line); }
    return line;
  } catch (FileNotFoundException e)
   {e.printStackTrace(); }
  return null; }
``` |

Table 3: Reading from a file: Outputs for the NSG and transformer baselines.

Instead, we consider four fidelity metrics: (1) *Set of API Calls*: Extract the set of API calls from the generated and reference codes, and compute the Jaccard similarity between the sets. (2) *Sequences of API Calls*: Generate the set of all possible API call sequences possible along code paths, and compute the Jaccard similarity between the sets for the generated and reference code. (3) *Sequences of Program Paths*: Generate the set of all possible paths from root to leaf in the AST, then compute the Jaccard similarity between the sets (two paths are equal if all elements except for object references match). (4) *AST Exact Match*: Exact AST match (except for object references), scored as 0 or 1. We compute the highest value for each metric across the ten bodies generated, and average the highest across all test programs. Results for these measures are presented in Table 2.

**Summary of results.** We find that in most cases, the NSG had a higher incidence of passing the various static checks compared to the baselines. This is perhaps not surprising, given that the NSG has access to the result of the static analysis via the attribute grammar. More intriguing is the much higher accuracy of the NSG for the fidelity results. Pre-trained language models and GNN2NAG are designed for next-token-prediction tasks (we give some results on these tasks in Appendix G). However, in our CPG task, no tokens are available from the method body to be generated. In particular, language models must treat the surrounding code and method header as input from which to generate the entire method body, and this proves difficult. The NSG, on the other hand, uses static analysis to symbolically extract this context, which is explicitly given to the neural network in the form of the class variables and methods that are available to be called (in $A(S)\downarrow$), and in the class name, encoded comments, variable names, and so on (in Z).

**Transformers vs. NSGs**: As a complement to our quantitative evaluation, we manually examined the outputs of our model and the baselines on a set of hand-written tasks for qualitative evaluation. The transformers produced impressively human-like code on several of these tasks. However, in quite a few cases, they produced incorrect programs that a trained human programmer would be unlikely to write. Also, the transformers were biased towards producing short programs, which often led them to produce uninteresting outputs.

Table 3 illustrates some of the failure modes of the transformer baselines. Here, we consider the task of reading a string from a file utility class. The top result for our NSG model declares a *String* variable to read from the already existing field while also correctly catching an *IOException*. The CODEGPT output in this case is unrelated to the context. CODEX initiates a FileReader object by invoking an argument which is of type *FileReader* itself, thereby causing a type mismatch. The code from GPT-NEO accesses a *file* instance variable that does not exist and also returns a blank line from the method. A few other examples of NSG and transformer outputs appear in Appendix A.

## 6   Related Work

**Non-Neural Models of Code.** Many non-neural models of code have been proposed over the years [25, 32, 28, 2, 27, 5]. A few of these models condition generation on symbolic information from the context. Specifically, Bielik et al. [5] use programmatically represented functions to gather

information about the context in which productions for program generation are fired, then utilize this information to impose a distribution on rules. Maddison & Tarlow [25] generate programs using a model that encodes a production's context using a set of "traversal variables." However, the absence of neural representations in these models puts a ceiling on their performance.

**Deep Models of Code.** There is, by now, a substantial literature on deep models trained on program syntax. Early work on this topic represented programs as sequences [32] or trees [26, 38, 9], and learned using classic neural models, such as RNNs, as well as specialized architectures [23, 30, 3]. The recent trend is to use transformers [36, 16, 14, 24]. Some of these models — for example, CODEGPT [24] — are trained purely on code corpora (spanning a variety of languages, including Java). Other models, such as CODEBERT [14], GPT-NEO [6], and CODEX [8], are trained on both natural language and code. In all of these cases, programs are generated without any explicit knowledge of program syntax or semantics.

The GNN2NAG model by Brockschmidt et al. [7] also uses an attribute grammar to direct the generation of programs. However, unlike our method, this model use a graph neural net to *learn* attributes of code. Our experiments show the benefits of our weak supervision approach over this.

Also related is work by Dai et al. [11], who extend grammar variational autoencoders [21] with hard constraints represented as attribute grammars. In that work, attribute constraints are propagated top-down, and every generated artifact is required to satisfy the top-level constraint. This strategy comes with challenges; as is well-known in the program-synthesis literature [31], top-down constraint propagation can lead to unsatisfiability, and require rejection of generated samples, for grammars above a certain level of complexity. We sidestep this issue by using attribute grammars as a form of weak supervision, rather than as a means to enforce hard constraints.

**Neurally Directed Program Synthesis.** Many recent papers study the problem of *neurally directed program synthesis* [4, 12, 34, 10, 33, 29]. Here, neural networks, and sometimes program analysis, are used to guide a combinatorial search over programs. Because such search is expensive, these methods are typically limited to constrained domain-specific languages. In contrast, our approach does not aim for a complete search over programs at generation time (our decoder does perform a beam search, but the width of this beam is limited). Instead, we embody our program generator as a neural network that sees program-analysis-derived facts as part of its data. This design choice makes our method more scalable and allows it to handle generation in a general-purpose language.

# 7 Conclusion

We have presented a framework for deep generation of source code in which the training procedure is weakly supervised by a static analyzer, in particular, an attribute grammar. We have shown that our implementation of this approach outperforms several larger, state-of-the-art transformers both in semantic properties and fidelity of generated method bodies.

A lesson of this work is that while modern transformers excel at writing superficially human-like code, they still lack the ability to learn the intricate semantics of general-purpose languages. At the same time, the semantics of code can be defined rigorously and partially extracted "for free" using program analysis. This extracted semantics can be used to aid neural models with the concepts that they struggle with during program generation. While we have used this idea to extend a tree LSTM, we could have implemented it on top of a modern transformer as well. We hope that future work will pursue such implementations.

Our work demonstrates an alternative use for formal language semantics, compared to how semantics are typically used in program synthesis research. Historically, semantics have been used to direct generation-time combinatorial searches over programs. However, scalability has been a challenge with such approaches. Our work points to an alternative use of semantics: rather than using semantic program analyses to direct a search over programs, one could use them to *annotate* programs at training and test time and leave the search to a modern neural network. We believe that such a strategy has the potential to vastly extend the capabilities of algorithms for program synthesis.

**Acknowledgments.** This work was funded by NSF grants 1918651 and 2033851; US Air Force and DARPA contract FA8750-20-C-0002; and ONR grants N00014-17-1-2889, N00014-19-1-2318, and N00014-20-1-2115. We thank Matt Amodio for his help with our initial exploration of the problem studied here, and Martin Rinard and the anonymous referees for their valuable comments.

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
