| | (a) Removing from a list | (b) Adding to a list |
|---|---|---|
| **Query** | ```
public class myClass{

  /** remove items from a list */
  public void
      remove(List<String> fp_2){
      }
}
``` | ```
public class myClass{

  /** add item to list */
  public void addItem (
  List<String> a, String b){
  }}
``` |
| **NSG** | ```
void remove(java.util.List
        <java.lang.Object> fp_2){
  java.util.Iterator var_8;
  var_8 = fp_2.iterator();
  while (var_8.hasNext()){
    java.lang.Object var_3;
    var_3 = new java.lang.Object(
      (java.lang.String) ARG,
      (javax.swing.Icon) ARG);
    var_3 = var_8.next();
    var_8.remove();
  }
  return;
}
``` | ```
public void addItem (
    List<String> fp_9,
    String fp_1) {
    if (fp_9.contains
          ((String) ARG)) {
        fp_9.add(fp_1);
    }
    return;
}
``` |
| **CODEGPT** | ```
public void remove(List<String> fp_2){
    items.remove(fp_2); }
``` | ```
public void addItem (
      List<String> a, String b)
  {
    items.add(a);
  }
``` |
| **CODEX** | ```
public void remove(List<String> fp_2){
    fp_2.remove(0); }
``` | ```
public void addItem (
  List<String> a, String b) {
  a.add(b); }
``` |
| **GptNeo1.3B** | ```
public void remove(List<String> fp_2){
    fp_2.remove(); }
``` | ```
public void addItem (
  List<String> a, String b) {
    List <String> temp =
        new ArrayList<String>();
  for (int i=0; i<a.size(); i++){
      temp.add(a.get(i));
  }
  a.add(b);
}
``` |

Table 4: Example synthesis outputs: (a) Removing from a list and (b) Adding to a list.

## A    Additional Synthesis Examples

Some additional program-generation examples are shown in Tables 4, 5, and 6.

## B    Static Checks Considered

Here we give an in-depth description of each of the static checks that has been tested in the paper as described in Section 5.2.

- **No Undeclared Variable Access.** All the variables used in a program should be declared before they are used and should be available in the scope. We measure the percentage of variable usages across all the programs that are declared before use. For example int bar() {x.write();} is a violation of this property because x is not declared (assuming that there is no field named x). When the statement x.write() is synthesized, the NSG-model has access to the symbol table at that point in the AST. The symbol table does not contain the variable x because it is not declared in the preceding statements. The NSG-model has learnt to use variables that are present in the symbol table (encoded as attribute "symTab"), which increases the likelihood that the variables used in the program are declared before being used.

- **Valid Formal Parameter Access.** All of the input variables accessed in a program body should be available in the class definition. Across all programs, we measure the percentage of input-variable accesses that are available. If a program grammar allows access for $n$ input variables, and the synthesizer tries to access one such variable $n_k$ even when it is not

| | (a) Writing to a file | (b) Creating a Swing JButton |
|---|---|---|
| **Query** | ```java
public class FileUtil{

  String err;
  public int read(File f){...}

  /* write lines to file */
  public void write(
    File f, String str){}}
``` | ```java
public class myGUI{

  public javax.swing.JFrame
      createFrame(final String title)
    {... }

  public void addActionListener
        (javax.swing.JButton title)
    {... }

  /**
  create a button
  */
  public
    javax.swing.JButton createButton(){ }
}
``` |
| **NSG** | ```java
void write(File f, String str){
  try {
    FileWriter var_0;
    var_0 = new FileWriter(f);
    var_0.write(str);
  } catch(IOException var_0) {
    var_0.printStackTrace();
    System.out.println( ARG ); }
  return; }
``` | ```java
javax.swing.JButton createButton(){
  javax.swing.ImageIcon var_9;
  var_9 = new javax.swing.ImageIcon.
                          ImageIcon(
          (java.lang.String) ARG,
          (java.lang.String) ARG
      );
  javax.swing.JButton var_7;
  var_7 = new javax.swing.JButton.JButton(
      java.lang.String: ARG,
      javax.swing.Icon: var_9);
  var_7.setVerticalTextPosition(
            (int) ARG);
  var_7.setHorizontalTextPosition(
            (int) ARG);
  return var_7;
}
``` |
| **CODEGPT** | ```java
void write (File a, String b){
System.out.println(a);
}
``` | ```java
public JButton createButton(){
    return new JButton(this);
}
``` |
| **CodeX** | ```java
public void write(File f, String str){
  Writer writer = null;
  try {
    writer = new BufferedWriter(
      new FileWriter(f));
    writer.write(str);
    writer.close();
  } catch (IOException e) {
    err = e.printStackTrace();}
}
``` | ```java
public JButton createButton(){
    final javax.swing.JButton b =
      new javax.swing.JButton();
    return b;
}
``` |
| **GptNeo1.3B** | ```java
public void write(File f, String str)
  { try { f.write(str); }
  catch ( IOException e) {
    e.printStackTrace(); }
}
``` | ```java
public JButton createButton(){
    javax.swing.JButton button =
      new javax.swing.JButton();
    button.addActionListener(this);
    return button;
}
``` |

Table 5: Example synthesis outputs: (a) Writing to a file and (b) Creating a Swing JButton. For brevity, we omit the method body in the query, denoted as **...**. The method body was fed into transformers as a part of the prompt during generation.

| | (c) Connect to a Socket | (d) Decrypting a Message |
|---|---|---|
| **Query** | ```java
public class tcpSocketManager{

  public Socket createSocket(final
    InetSocketAddress socketAddress){
    ...
  }

  public void connect(
    InetSocketAddress socketAddress){
  }
}
``` | ```java
public class lockAndKey{

 public String encryptMessage(
    String message, String secret_key){
    ...
 }

 public String decryptMessage(
    String encrypted_message,
    String secret_key){
 }
}
``` |
| **NSG** | ```java
public void connect(
    InetSocketAddress fp_3)
    Socket var_8;
    var_8 = new Socket.Socket(
      (String) ARG,(int) ARG);
    try {
      InetSocketAddress var_3;
      var_3 = new InetSocketAddress.
        InetSocketAddress((String) ARG,
                     (int) ARG);
      var_8.connect((SocketAddress) var_3,
                (int) ARG);
    }
    catch(ConnectException var_3){
        var_3.printStackTrace();
    }
    try {
        var_8.close();
    }
    catch(IOException var_3){
        var_3.printStackTrace();
    }
    return;
``` | ```java
String decryptMessage(String fp_2,
                      String fp_5){
  if (fp_5.equals((Object) ARG)){
    return fp_5;
  }
  int var_6;
  var_6 = fp_5.indexOf((String) fp_5);
  if (true){
    return fp_5;
  }
  String var_4;
  var_4 = fp_5.substring((int) ARG,
                    (int) var_6);
  return var_4;
}
``` |
| **CODEGPT** | ```java
public void connect(InetSocketAddress
                    socketAddress)
  if (socketAddress ~= null) {
  socketAddress.connect(socketAddress);
}
``` | ```java
public String decryptMessage(
         String encrypted_message,
         String secret_key){
return decryptMessage(encrypted_message,
                   secret_key);
}
``` |
| **CODEX** | ```java
public void connect(InetSocketAddress
                    socketAddress){
  AbstractSocket activeProxySocket =
    createSocket(socketAddress);
  activeProxySocket.connect(
    socketAddress);
}
``` | ```java
public String decryptMessage(
         String encrypted_message,
         String secret_key){
 Key aesKey = new SecretKeySpec(
   secret_key.getBytes(), 'AES');
 Cipher cipher =
   Cipher.getInstance('AES');
 cipher.init(Cipher.DECRYPT_MODE, aesKey);
 return cipher.doFinal(encrypted_message);
}
``` |
| **GptNeo1.3B** | ```java
public void connect(InetSocketAddress
                    socketAddress)
  throws IOException {
    Socket s = new Socket(
      new Proxy(
        Proxy.Type.SOCKS,socketAddress)
      );
    s.connect();
}
``` | ```java
public String decryptMessage(
         String encrypted_message,
         String secret_key){
 Key aesKey=
   new SecretKeySpec(
     encrypted_message.getBytes(),
     secret_key.getBytes()
   );
   return aesKey.doFinal(
     encrypted_message);
}
``` |

Table 6: Example synthesis outputs: (a) Connect to a socket and (b) Decrypting a message. For brevity, we omit the method body in the query, denoted as **...**. The method body was fed into transformers as a part of the prompt during generation.

available, this property will be violated. The formal-parameter-type information is present in *symTab* corresponding to each of the input variable which helps NSG learn this property correctly.

- **Valid Class Variable Access.** All the class fields accessed in a program body should be available in the class definition. The presence of field information in *symtab* helps NSG satisfy this semantic property. Across all programs, we measure the percentage of field accesses that happened when they were available.

- **No Uninitialized Objects.** All variables with reference type should be initialized. Out of all the variables with reference types declared across all the programs, we measure the percentage of variables that are initialized using a "`new`" statement. For example, `BufferedWriter x; x.write();` is a violation because `x` is not initialized using `new BufferedWriter`. Violation of this property could cause a *NullPointerException* at runtime. The AG keeps track of variable initializations using an attribute of type array of Booleans, named *IsInitialized*. Whenever a variable is declared, the corresponding value in the *IsInitialized* array is set to *False*. As soon as the variable is initialized, the attribute is set to *True*. This attribute helps the NSG model learn to avoid generating method bodies in which a variable is used without being initialized.

- **No Variable Access Errors.** This property is the aggregate of the preceding four semantic checks.

- **Object-method compatibility.** Methods should be invoked on objects of appropriate types. For example, consider the program snippet `int k = 0;`
`bool b = k.startsWith(pre);` which contains an invocation of `String::startsWith(String pre)`. This program fails the object-method-compatibility check because the method is invoked on a variable of type *int* instead of a variable of type *String*. The method invocation *startsWith* is synthesized before the terms *k* and *pre*.[2] The symbol-table attribute of the AG, in combination with the synthesized attribute *expr_type*, helps the NSG model avoid such cases by learning to synthesize an expression with a compatible type, given the method signature to be invoked on it.

- **Return Type at the Call Site.** The return type of a method invoked at some call site should be consistent with the type expected at the call site. In `bool b = aStr.startsWith(pre);` this property asserts that the type of `b` should be compatible with the return type of `String::startsWith`. The symbol table alongside the synthesized attribute from the API call, namely *ret_type*, helps the NSG respect this rule.

- **Actual Parameter Type.** The actual-parameter types in an API call should be consistent with the corresponding formal-parameter types. For example, consider the program fragrment `int pre = 0; bool b = aStr.startsWith(pre);` which contains an invocation of `String::startsWith(String pre)`. This program fails the formal-parameter type check because the actual-parameter type (*int*) does not match the formal-parameter type (*String*). The AG has the symbol-table attribute, which contains the variables in scope and their types, plus it has access to the intended type from the API call by the attribute *typeList*, which helps the NSG model to learn to synthesize API arguments of appropriate types.

- **Return Statement Type.** The type of the expression in a return statement of a method should be consistent with the method's declared return type. For example, `public int foo(){String x; return x}` violates this property because the returned expression `x` is of type *String*, whereas the declared return type is *int*. To make it easier for the NSG model to learn this restriction, the AG has a dedicated attribute *methodRetType* to propagate the declared return type throughout the method, which helps it generate an expression of the appropriate type after consulting the symbol table. For this property, we measure the

---

[2]Note that while this attribute grammar requires *Method* to be expanded before *Expr* (because inherited attributes of the latter depend on synthesized attributes of the former), the grammar is still *L*-attributed if we expand *Method* then *Expr*, and perform an unparsing trick to emit the subtree produced by *Expr* first.

percentage of return statements for which the expression type matches the method's declared return type.

- **No Type Errors.** All variables should be accessed in a type-consistent manner. Across all the programs, we measure the percentage of variable accesses that are type-consistent. Locations of variable accesses include all types of variable accesses relevant to an API call, method arguments, return statements, the variables on which the methods are invoked, variable assignments, and internal class method calls.

- **Return Statement Exists.** This property asserts that a method body should have a return statement. `public int foo(){String x;}` violates this property because the method body does not have a return statement. The AG propagates an attribute, *retStmtGenerated*. This attribute is initially set to *false*. When a return statement is encountered, the attribute is set to *true*. The NSG model learns to continue generating statements while this attribute is *false*, and to stop generating statements in the current scope when the attribute is *true*. For this property, we report the percentage of programs synthesized with a `return` statement.

- **No Unused Variables.** There should not be any unused variables (variables declared but not used) in the method body. For example, in `public void reader(){String x; String y; x=field1.read()}`, the variable y is an unused variable. To keep track of the unused variables, we use a boolean array attribute *isUsed*. Entries in array corresponding to the used variables are *true* whereas all other entries are *false*. Out of all the programs synthesized, we report the percentage of variables declared which have been used inside the method body.

- **Percentage of Parsing.** A parser for the Java language should be able to parse the synthesized programs. We use an open-source Java parser, called javalang [19], and check for the number of programs that parse. This test does not include static-semantic checks; it only checks if a generated program has legal Java syntax. Note that NSG, CNG, and GNN2NAG models are rule-based generation and they are bound to parse by definition. The pre-trained language models, however, are not guaranteed to produce programs that exactly follow the grammar definition. Therefore we capture all such instances that throw parsing exceptions and report the resulting numbers.

- **Pass All Checks.** This property is the aggregate of all of the preceding checks.

## C   Implementation Details

We now give a few details about how the distributions required to instantiate an NSG are implemented in our Java prototype.

**Evidence Encoder**: The evidences that we support as input to user context include class-level information (e.g., class name, Java-type information of the instance variables, and methods in the same class that have already been implemented); along with the information from the method header.

Each of these evidence types is encoded in a way appropriate to its domain. The method header has a separate encoding for each of its components: return type, formal parameters, and method name. The natural-language description available as Javadoc is also included. In total, there are seven kinds of evidence that we consider in our context.

The evidences are encoded together as follows: class and method names are split using camel case and delimiters; the resulting elements are treated as natural-language keywords, and encoded as a set, using a single-layer feed-forward network. The other evidences that are represented as sets and encoded by similar neural network. The type information of the class variables, formal parameters, and Javadoc are encoded as sequential data using a single-layered LSTM. The surrounding method is encoded as a concatenation of the three components of the method header, namely, the method name, formal parameters, and return type, followed by a dense layer to reduce the dimensionality to the size of the latent space. Note that the model defined in Section 4 allows us to get meaningful synthesis outputs even when only a small subset of the different kinds of evidence are available during training.

**Sampling Symbol RHS Values**: The distribution $P(S|\mathsf{SymSoFar}, A(S)\downarrow, \mathsf{Z})$ is implemented using an LSTM. There are six different kind of symbols for which we need to chooses an RHS: choosing the program block to produce (e.g., producing a try-catch statement or a loop), Java types, object-initialization calls, API calls, variable accesses, and accessing a method within the same class. Each one of these has their own vocabulary of possibilities, and requires a separate neural LSTM decoding unit. It is also possible to use additional, separate neural units in different production scenarios. In our implementation, we use four separate LSTM units for decoding variable accesses: for a variable that is being declared, when accessed in a return statement, when accessed as an input parameter, or when an API call is invoked. In other words, the NSG synthesizer consists of multiple decoding neural units, for decoding all of the production rules in the program's grammar, each using a separate LSTM unit. It should be noted here that even though each of these LSTM units in the network has its own parameter set, they all maintain the same recurrent state, which tracks the state of the unfinished program synthesized so far.

**Attributes**: Each of the neural units in an NSG decodes the current symbol using its corresponding LSTM and additional attributes available from the attribute grammar. Generally when a recurrent model like an LSTM is trained, the input to the LSTM cell is fixed as the correct output from the last time step (or the output from the parent node in case of a tree decoder). The availability of attributes in an NSG lets us augment this input information with the additional attributes from our grammar. The attributes that we support are given below:

- Symbol table: An unrolled floating-point matrix that represents the types of all variables in scope, including field variables, input variables, and user-defined variables. Represented in our grammar as *symTab* attrbute.

- Method return type: A floating-point vector containing the expected type of the method body. Represented in our grammar as *methodReturnType*.

- Return type of an API call, expression type of an object invoking an API call, and types of the input variables of an API call: Three separate floating-point vectors that represent the expected return type (*retType*), the expression type of the object that initiates an API call (*exprType*), and the expected formal parameters of the API call, if any (*typeList*).

- Internal-method table: A floating-point vector representing the neural representation of the completed methods available in the same class.

- Unused variable flag: A Boolean vector indicating which variables have been initialized but not used so far in the program. The attribute that tracks this semantic property is *isUsed*.

- Uninitiated-object flag: A Boolean vector indicating which objects have been declared but not initialized. The attribute that tracks this semantic property is *isInitialized*.

- Return-statement flag: A Boolean indicating if a return statement has yet been reached in the program body. The attribute that tracks this semantic property is *retStmtGenerated*

Note that not all attributes are important to every production rule in the grammar at a given time step. For example, while decoding a variable access, it is unimportant to know about internal methods. This information follows from our attribute grammar, as described in Appendix K. If a particular attribute is not associated with a non-terminal, or it is unused inside a production rule, it is not required required for that rule, and the attribute can be omitted from being input to decoding that particular token.

**Training and Inference**: During training, all information required for the LSTM, such as contextual information for the missing class and the associated attributes in the AST, are available to the neural decoder. The neural network is only tasked with learning the correct distributions related to decoding the method-body AST. The objective function related to learning the probability distributions within the learner is composed of a linear sum of cross-entropy loss for each category of symbol: non-terminals of the AST, API calls, Java types, and so on. This loss can be minimized using standard techniques like gradient descent to 'train' the model. If we compare this to a simple neural model, the NSG decoder has additional inputs in the form of attributes coming from the grammar to aid its decoding process.

During inference, only the program context is available to the decoder. The attributes that the synthesizer requires are inferred on-the-fly, after the previous sub-part of the AST has been decoded. Because we make use of an L-attributed grammar, at each step of AST construction, the necessary

inputs are already available from the partial AST at hand. At no point in the decoding process do the neural units need any input from the part of the AST that has not yet been synthesized. This approach to synthesizing is close to the standard inference procedure in sequence-to-sequence models [35, 13]

Given the learned neural units, decoding for the "best" program is an intractable problem, hence we use beam search [37]. Because beam search is only applicable for sequences, we modify it to perform a depth-first traversal on the AST, with the non-terminal nodes that lead to branching in the AST stored in a separate stack.

# D   Implementation of Baselines

**GNN2NAG**: We describe the implementation details of GNN2NAG [7], which is one of the baselines in Sec. 5. We expand the AST nodes in the order of a depth-first search. We considered the same six types of edges as Brockschmidt et al. [7], which consists of *Parents*, *Child*, *NextSib*, *NextUse*, *NextToken*, and *InhToSyn*. After building the graph, we propagate the information through a Gated Graph Neural Network (GGNN [22, 7]). We obtain a representation for each node after the GGNN propagation. We then apply an LSTM to go over all the nodes until reaching the node where the goal is to predict the next token. Training is via cross-entropy loss. Note that the biggest difference between our implementation of GNN2NAG and Brockschmidt et al. [7] is the use of an LSTM. Brockschmidt et al. [7] assumes that information about which type of edge is responsible for generating the token is available to the model. However, this information is not available in our setup. Thus, we use an LSTM to iterate over all edges in the GNN to obtain the features for prediction.

**Pre-Trained Language Models**: We consider 4 types of transformer models—GPTNeo 125M, GPT-Neo 1.3B, CODEGPT, and CODEX [6, 24, 8]. We fine-tuned each of these pre-trained transformers on our Java dataset, except for CODEX, for which we have no access to the pre-trained weights. While our NSG model only takes the headers in the Java class as inputs, for the various transformer models, the input is the entire Java class, including both headers and method bodies. (We found that transformers perform quite poorly if only headers are provided.) During evaluation, transformers are asked to predict the missing method body given the header and the rest of the class.

To fine-tune on our Java dataset, we used the token-level code-completion task provided by CodeXGLUE[3] [24]. During fine-tuning, the transformers are asked to predict the next token in an autoregressive fashion, just like in any language-modeling task. The learning rate is $8e{-}5$ and the batch size is 1 on each GPU. In total, we used 16 GPUs to fine-tune these transformers, which takes about 4 days to complete two epochs on our Java dataset, consisting of 600,000 training instances.

# E   Generation of Training Data

Now we sketch the process by which our training data is generated. Assume that the task is to generate procedure bodies from start-nonterminal *Prog*, and that we are given a large corpus of Java programs from which to learn the distribution $P(Prog|\mathsf{X})$.

An AG-based compiler is used to produce the training data. For each user-defined method $M$ in the corpus, we create training examples of the form

$$\left((Prog, \mathsf{S}_1^{\mathrm{rhs}}), ..., (S_{i-1}, \mathsf{S}_{i-1}^{\mathrm{rhs}}), (S_i, \mathsf{S}_i^{\mathrm{rhs}}), A(S_i)\!\downarrow, \mathsf{X}\right) \tag{2}$$

where (i) $(Prog, \mathsf{S}_1^{\mathrm{rhs}}), ..., (S_{i-1}, \mathsf{S}_{i-1}^{\mathrm{rhs}})$ is a pre-order listing—from goal nonterminal *Prog* to a particular instance of nonterminal $S_i$—of the (nonterminal, RHS) choices in $M$'s *Prog* subtree, (ii) $\mathsf{S}_i^{\mathrm{rhs}}$ is the RHS production that occurs at $S_i$, and (iii) attribute values $A(S_i)\!\downarrow$ are the values at the given instance of $S_i$. As input-output pairs for a learner, inputs (i) and (iii) produce output (ii).

We compile the program, create its parse tree, and label each node with the values of its attributes (which are evaluated during a left-to-right pass over the tree). For each method $M$, its subtree is traversed, and a training example is emitted for each node of the subtree.

---

[3] https://github.com/microsoft/CodeXGLUE/tree/main/Code-Code/CodeCompletion-token

Table 7: Percent of Static Checks Passed with 25% Evidence

|  | GPTNeo125M | GPTNeo1.3B | CodeX | CODEGPT | GNN2NAG | CNG | NSG |
|---|---|---|---|---|---|---|---|
| No Undeclared Variable Access | 90.77% | 89.99% | 84.55% | 89.21% | 46.88% | 19.78% | **99.32%** |
| Valid Formal Param Access | NA | NA | NA | NA | 25.72% | 11.03% | **98.61%** |
| Valid Class Var Access | NA | NA | NA | NA | 14.34% | 12.75% | **99.31%** |
| No Uninitialized Objects | 92.35% | 91.21% | 88.52% | **93.40%** | 20.31% | 21.56% | 93.35% |
| No Variable Access Error | 90.92% | 90.11% | 84.98% | 89.63% | 28.10% | 17.92% | **99.10%** |
| Object-Method Compatibility | 96.93% | 97.05% | 96.74% | **98.35%** | 21.29% | 12.23% | 94.87% |
| Ret Type at Call Site | 97.91% | 97.66% | **98.47%** | 97.98% | 22.97% | 16.40% | 92.53% |
| Actual Param Type | 88.51% | 88.61% | 90.02% | 86.77% | 9.22% | 16.09% | **93.63%** |
| Return Stmt Type | 82.50% | 81.56% | 83.75% | 83.43% | 12.05% | 9.51% | **88.94%** |
| No Type Errors | 86.46% | 86.14% | 88.62% | 86.83% | 15.98% | 13.56% | **91.98%** |
| Return Stmt Exists | 99.58% | 99.61% | 96.74% | 99.56% | 93.83% | **99.92%** | 96.94% |
| No Unused Variables | 96.51% | 96.33% | 96.46% | **97.60%** | 20.14% | 24.29% | 91.75% |
| Percentage of Parsing | 98.61% | 98.53% | 94.95% | 97.14% | **100.0%** | **100.0%** | **100.0%** |
| Pass All Checks | 65.26% | 62.65% | 38.77% | 63.12% | 16.75% | 12.87% | **79.17%** |

Table 8: Average Fidelity of Generated Method Bodies with 25% Evidence

|  | GPTNeo125M | GPTNeo1.3B | CodeX | CODEGPT | CNG | NSG |
|---|---|---|---|---|---|---|
| Set of API Calls | 24% | 27% | 29% | 27% | 12% | **43%** |
| Sequences of API Calls | 12% | 14% | 13% | 13% | 7% | **31%** |
| Sequences of Program Paths | 7% | 8% | 8% | 8% | 7% | **28%** |
| AST Exact Match | 7% | 8% | 8% | 8% | 1% | **18%** |

# F   Restricting Available Evidence

In our experiments, generation of a particular method is conditioned on available "evidences," which refer to the context surrounding the missing method, in the method's complete class (other method names and method headers, Java Doc comments, class variables, and so on). All of the experiments described thus far simulate the situation where the entire class—except the method to be generated—is visible when it is time to generate the missing method. This simulates the situation where a user is using an automatic programming tool to help generate the very last method in a class, when all other methods and class variables have been defined and are visible.

We can restrict the amount of evidence available to make the task more difficult. When we only make a portion of the evidence available, this simulates the case where a user is using an automatic programming tool to generate a method when the surrounding class is less complete. When we use "$x$% evidence" for a task, each piece of evidence in the surrounding code is selected and available to the automatic programming tool with $x$% probability. In Table 7 and Table 8, we show results obtained when we repeat the experiments from earlier in the paper, but this time using 25% evidence while Table 9 and Table 10 show the result for 50% evidence.

# G   Next-Token Prediction

Our NSG implementation uses a relatively weak language model (based on LSTMs as opposed to more modern transformers) but augments them with a static analysis. We have shown that the resulting NSG is good at "long-horizon" tasks such as semantic consistency (compared to the baselines tested) and at generating methods that have high fidelity to the original, "correct" method. But it is reasonable to ask: how does the NSG compare to the baselines at "short-horizon" tasks? To measure this, for each symbol $S$ that is expanded to form the body of a test method, we compute (i) the actual left-context sequence of the test method (up to but not including the RHS sequence chosen for $S$) as the value of SymSoFar, (ii) $A(S)\downarrow$ (in the case of the NSG), and (iii) Z. We then use these values to ask the NSG to predict the next RHS. If the predicted RHS matched the observed RHS, the model was scored as "correct." We recorded the percentage of correct predictions for terminal RHS symbols (such as API calls or types) for each test program.

We also performed next-token prediction using the NSG and three of the baseline models. Note that it is non-trivial to classify CODEGPT's output into different terminal symbols, so we only report the overall RHS symbols' correctness. The results show that two of the baselines (CODEGPT and GNN2NAG) are very accurate, and demonstrate better performance than the NSG on this task.

Table 9: Percent of Static Checks Passed with 50% Evidence

|  | GPTNeo125M | GPTNeo1.3B | CodeX | CODEGPT | GNN2NAG | CNG | NSG |
|---|---|---|---|---|---|---|---|
| No Undeclared Variable Access | 89.87% | 90.36% | 88.62% | 90.34% | 47.17% | 17.79% | **99.86%** |
| Valid Formal Param Access | NA | NA | NA | NA | 25.50% | 8.58% | **99.83%** |
| Valid Class Var Access | NA | NA | NA | NA | 14.96% | 11.57% | **99.78%** |
| No Uninitialized Objects | 93.90% | 91.73% | 90.82% | 94.37% | 20.01% | 21.68% | **97.30%** |
| No Variable Access Error | 90.36% | 90.51% | 88.86% | 91.32% | 28.43% | 17.34% | **99.84%** |
| Object-Method Compatibility | **98.36%** | 98.09% | 98.35% | 97.84% | 21.39% | 10.11% | 96.42% |
| Ret Type at Call Site | 97.38% | 98.01% | **98.53%** | 97.83% | 23.45% | 14.82% | 97.22% |
| Actual Param Type | 87.03% | 86.36% | 92.28% | 88.71% | 9.24% | 14.35% | **96.74%** |
| Return Stmt Type | 84.05% | 85.09% | 88.13% | 85.23% | 12.07% | 7.66% | **92.15%** |
| No Type Errors | 87.25% | 88.13% | 91.42% | 88.10% | 16.04% | 11.45% | **96.22%** |
| Return Stmt Exists | 99.61% | 99.80% | 98.44% | **99.57%** | 93.87% | 98.71% | 97.47% |
| No Unused Variables | 96.42% | 96.46% | 96.82% | **97.64%** | 20.55% | 18.50% | 94.20% |
| Percentage of Parsing | 98.18% | 98.13% | 94.69% | 97.08% | **100.0%** | **100.0%** | **100.0%** |
| Pass All Checks | 65.26% | 64.88% | 47.49% | 67.73% | 16.92% | 24.28% | **86.00%** |

Table 10: Average Fidelity of Generated Method Bodies with 50% Evidence

|  | GPTNeo125M | GPTNeo1.3B | CodeX | CODEGPT | CNG | NSG |
|---|---|---|---|---|---|---|
| Set of API Calls | 32% | 37% | 36% | 36% | 12% | **50%** |
| Sequences of API Calls | 17% | 20% | 16% | 19% | 7% | **39%** |
| Sequences of Program Paths | 13% | 10% | 10% | 14% | 7% | **36%** |
| AST Exact Match | 13% | 10% | 10% | 14% | 1% | **21%** |

These results are in-keeping with our assertion that the baselines are useful mostly for short-horizon code-generation tasks. However, they struggle with long-horizon tasks, such as the CPG task of generating an entire Java method body. The results—together our earlier CPG results—also show that even though the NSG has reduced accuracy in a short-horizon task, it is still able to generate semantically accurate programs on the CPG task.

# H    Application to Novel Semantic Checks

The NSG approach can generate semantically accurate programs given context. At its core, an NSG relies on the various semantic properties (i.e., attributes) on which it is trained. We would like to understand the influence of these semantic properties in the generated program, and explore the possibility that training on such a set of attributes can automatically allow for high accuracy with respect to additional semantic checks for which specific attributes were not explicitly provided during training. To study this question, we performed an ablation study in which we trained an NSG with a subset of the relevant attributes, but evaluated the generated programs on all properties.

We trained an NSG without the *attrOut.retStmtGenerated* and *methodRetType* attributes, as defined in Section K.2. With 50% of the evidence available, we see that the resulting model suffers in terms of accuracy. The "Return Stmt Type" accuracy falls from 92.15% to 77.45% whereas the "Return Stmt Exists" accuracy falls from 97.47% to 95.68%. That said, note that the resulting "Return Stmt Type" accuracy is still a big improvement over the vanilla CNG model (with no attributes), which is correct only 9.51% of the time.

This suggests that the NSG has learned type-safe program generation from other semantic properties, most notably the *symTab* attribute, which carries type information about the various objects that are currently in scope. This further suggests that providing a small core of key attributes may be enough to greatly increase the accuracy of code generation.

# I    Robustness Incomplete Analysis

In this section, we analyze a situation where the static analyzer fails to accurately resolve different attributes during the synthesis process. We simulate three situations in which the static analyzer might fail.

In the first scenario, we emulate a situation where the compiler is unable to resolve the correct return-type information from the missing method that the user has asked the NSG to synthesize. This

Table 11: Next-Token Prediction Accuracy

| | Percentage of Evidence Available | | | | | | | |
| | 50% | | | | 100% | | | |
| | NSG | CODEGPT | GNN2NAG | CNG | NSG | CODEGPT | GNN2NAG | CNG |
|---|---|---|---|---|---|---|---|---|
| API Calls | 62.42% | NA | **80.24%** | 49.05% | 75.94% | NA | **80.77%** | 59.73% |
| Object Initialization Call | 59.64% | NA | **97.65%** | 49.12% | 66.66% | NA | **97.94%** | 87.90% |
| Types | **61.11%** | NA | 85.78% | 50.28% | 70.33% | NA | **86.21%** | 54.44% |
| Variable Access | **92.26%** | NA | 92.11% | 50.28% | 92.44% | NA | **92.94%** | 52.85% |
| All Terminal RHS Symbols | 73.41% | **88%** | 80.83% | 51.22% | 73.99% | **89%** | 81.1% | 54.32% |

| Real Code | CodeGPT | NSG |
|---|---|---|
| ```
public String reader()
{
  StringBuffer stringBuffer
        = new StringBuffer();
  String line;
  while ((line =
  bReader.readLine() ~= null) {
    stringBuffer.append(line);
    stringBuffer.append("\n");};
  return stringBuffer.toString();
}
``` | ```
public String reader()
{
  StringBuffer buffer=
       new StringBuffer();
  buffer.append("\n");
  return buffer.toString();
}
``` | ```
public String reader()
{
  java.lang.String var_9;
  try{
   var_9=field_5.readLine();
   }
  catch(IOException var_8) {
   var_8.printStackTrace();
  }
  return var_9;
}
``` |

Table 12: Reader example for analyzing the BLEU-score metric.

results in a default `null` value being passed around for the attribute *methodRetType*. We find that this reduces the overall accuracy for the attribute "Return Stmt Type" from 90.97% to 77.28%. This does not seem to impact other static checks, however.

In the second scenario, consider the case where the compiler is unable to resolve the API return-type attribute *retType*. This reduces the accuracy of the "Return type at call site" check from 98.01% to 18.16%. It also results in a decrease in "No undeclared-variable access" and "Valid formal-param access" to 72.48% and 67.23%, respectively. This is is a huge decrease from 99.82% and 99.55% accuracy that these semantic checks had for the base model where the attribute *retType* can be resolved correctly. The fidelity metrics are also impacted, where the "AST exact match" metric drops from 26% to 10%. This is because the *retType* attribute is used in many portions of our attribute grammar, on which the trained program generator is being conditioned on. An incorrect resolution of such attribute had led to deterioration of the overall model performance.

In the final scenario, we only break the static analyzer's capability to resolve the unused-variable-check attribute *attrOut.isUsed*. For this scenario, we see that only the one semantic check "No unused variables" out of all the semantic checks considered is impacted. Here the accuracy for this check drops from 93.84% to 91.10%. All other metrics have negligible changes.

Rather unsurprisingly, the results suggest that NSG relies heavily on the static analyzer, and that some attributes influence the result much more than others. It is also critical to have a static analyzer that performs accurately during inference time, to avoid any model performance degradation.

## J  BLEU-Score Analysis

As described in the main body of the paper, BiLingual Evaluation Understudy or BLEU score is "problematic in the code-generation setting. First, the BLEU score is not invariant to variable renamings, which means that a nonsensical program that uses commonplace variable names can get an artificially high BLEU score. Second, programs are structured objects in which some tokens indicate control flow, and some indicate data flow. The BLEU score does not take this structure into account." We use one of the examples in Table 4 of the paper ("reading from a file") to illustrate this point and show the real code and outputs from CodeGPT and NSG in Table 12.

The NSG output is clearly better. However, the CodeGPT output gets a higher BLEU score because it uses variable names that superficially match the ground truth. Specifically, the BLEU score of the CodeGPT output is 25.11 and the BLEU score of the NSG output is 19.07. This situation arose often in our experiments, which is why we have used alternative program-equivalence metrics to judge performance of generated programs, as defined in Section 5.

# K Grammar

The Neural Attribute Grammar (NSG) model learns to synthesize real-life Java programs while learning over production rules of an attribute grammar. In this section, we present the comprehensive set of production rules considered, along with the attributes used. We first present the context-free grammar in Appendix K.1, and then decorate it with attributes in Appendix K.2. The productions in **a-c** deal with expansion of all the non-terminal symbols in the grammar: rules in **a** mainly expand to one line of code in a Java method body; rules in **b** are their corresponding expansions; and rules in **c** deal with control-flow operations inside the grammar. Rules in **d** generate terminal symbols inside the grammar. We show the flow of attributes *symTab* and *methodRetType* in the AST in Appendix K.2. The rest of the attributes are passed inside *attrIn* and *attrOut*, namely *isInitialized*, *isUsed*, *retStmtGenerated* and *itrVec*.

## K.1 Context Free Grammar

**a1.** *Start* : *Stmt*
**a2.** *Stmt* : *Stmt* ; *Stmt* | $\epsilon$
**a3.** *Stmt* : *Decl*
**a4.** *Stmt* : *ObjInit*
**a5.** *Stmt* : *Invoke*
**a6.** *Stmt* : *Return*

**b1.** *Decl* : *Type Var*
**b2.** *ObjInit* : *Type Var* **= new** *Type ArgList*
**b3.** *Invoke* : *Var* = *Var Call InvokeMore*
**b4.** *InvokeMore* : *Call InvokeMore* | $\epsilon$
**b5.** *Call* : *Api ArgList*
**b6.** *ArgList* : *Var ArgList* | $\epsilon$
**b7.** *Return* : **return** *Var*

**c1.** *Stmt* : *Branch* | *Loop* | *Except*
**c2.** *Branch* : **if** *Cond* **then** *Stmt* **else** *Stmt*
**c3.** *Loop* : **while** *Cond* **then** *Stmt*
**c4.** *Except* : **try** *Stmt Catch*
**c5.** *Catch* : **catch**(*Type*) *Stmt*; *Catch* | $\epsilon$
**c6.** *Cond* : *Call*

**d1.** *Api* : **JAVA_API_CALL**
**d2.** *Api* : **INTERNAL_METHOD_CALL**
**d3.** *Type* : **JAVA_TYPE**
**d4.** *Var* : **VAR_ID**

## K.2 Attribute Grammar

**a0.** Initialization of inherited attributes of *Start*:
$\big[$ *Start*.symTab $\downarrow$ :=
$\{$ $in\_param\_1 \mapsto type\_in\_param\_1$,
$\dots$
$in\_param\_n \mapsto type\_in\_param\_n$,
$field\_1 \mapsto type\_field\_1$,
$\dots$
$field\_m \mapsto type\_field\_m$ $\}$
*Start*.attrIn.itrVec $\downarrow$ := (false, false);
*Start*.attrIn.retStmtGenerated $\downarrow$ := false;
*Start*.attrIn.isInitialized $\downarrow$ := $\phi$;
*Start*.attrIn.isUsed $\downarrow$ := $\phi$;
*Start*.methodRetType $\downarrow$ := $METHOD\_RET\_TYPE$; $\big]$

**a1.** *Start* : *Stmt* ;
$\big[$ *Stmt*.attrIn ↓ := *Start*.attrIn ↓;
*Stmt*.methodRetType ↓ := *Start*.methodRetType ↓;
*Stmt*.symTab ↓ := *Start*.symTab ↓;
*Start*.symTabOut ↑ := *Stmt*.symTabOut ↑;
*Start*.attrOut ↑ := *Stmt*.attrOut ↑;
*Start*.valid ↑ := *Stmt*.valid ↑; $\big]$

**a2a.** *Stmt*$0 : *Stmt*$1 ; *Stmt*$2
$\big[$ *Stmt*$1.symTab ↓ := *Stmt*$0.symTab ↓;
*Stmt*$2.symTab ↓ := *Stmt*$1.symTabOut ↑;
*Stmt*$0.symTabOut ↑ := *Stmt*$2.symTabOut ↑;
*Stmt*$1.attrIn ↓ := *Stmt*$0.attrIn ↓;
*Stmt*$2.attrIn ↓ := *Stmt*$1.attrOut ↑;
*Stmt*$0.attrOut ↑ := *Stmt*$2.attrOut ↑;
*Stmt*$1.methodRetType ↓ := *Stmt*$0.methodRetType ↓;
*Stmt*$2.methodRetType ↓ := *Stmt*$0.methodRetType ↓;
*Stmt*$0.valid ↑ := *Stmt*$1.valid ↑ ∧ *Stmt*$2.valid ↑; $\big]$

**a2b.** *Stmt* : $\epsilon$
$\big[$ *Stmt*.symTabOut ↑ := {};
*Stmt*.attrOut.itrVec ↑ := (false, false);
*Stmt*.valid ↑ := true; $\big]$

**a3.** *Stmt* : *Decl*
$\big[$ *Decl*.symTab ↓ := *Stmt*.symTab ↓;
*Stmt*.symTabOut ↑ := *Stmt*.symTab ↓ + *Decl*.symTabOut ↑;
*Decl*.attrIn ↓ := *Stmt*.attrIn ↓;
*Stmt*.attrOut ↑ := *Stmt*.attrIn ↓ + *Decl*.attrOut ↑;
*Decl*.methodRetType ↓ := *Stmt*.methodRetType ↓;
*Stmt*.valid ↑ := *Decl*.valid ↑; $\big]$

**a4.** *Stmt* : *ObjInit*
$\big[$ *ObjInit*.symTab ↓ := *Stmt*.symTab ↓;
*Stmt*.symTabOut ↑ := *Stmt*.symTab ↓
            + *ObjInit*.symTabOut ↑;
*ObjInit*.attrIn ↓ := *Stmt*.attrIn ↓;
*ObjInit*.methodRetType ↓ := *Stmt*.methodRetType ↓;
*Stmt*.attrOut ↑ := *Stmt*.attrIn ↓ + *ObjInit*.attrOut ↑;
*Stmt*.valid ↑ := *ObjInit*.valid ↑ $\big]$

**a5.** *Stmt* : *Invoke*
$\big[$ *Invoke*.symTab ↓ := *Stmt*.symTab ↓;
*Stmt*.symTabOut ↑ := *Stmt*.symTab ↓ + *Invoke*.symTabOut ↑;
*Invoke*.attrIn ↓ := *Stmt*.attrIn ↓;
*Stmt*.attrOut ↑ := *Stmt*.attrOut ↓ + *Invoke*.attrOut ↑;
*Invoke*.methodRetType ↓ := *Stmt*.methodRetType ↓;
*Stmt*.valid ↑ := *Invoke*.valid ↑; $\big]$

**a6.** *Stmt* : *Return*
$\big[$ *Return*.symTab ↓ := *Stmt*.symTab ↓;
*Stmt*.symTabOut ↑ :=
    *Stmt*.symTab ↓ + *Return*.symTabOut ↑;
*Invoke*.attrIn ↓ := *Stmt*.attrIn ↓;
*Stmt*.attrOut ↑ := *Stmt*.attrIn ↓ + *Invoke*.attrOut ↑; $\big]$
*Return*.methodRetType ↓ := *Stmt*.methodRetType ↓;
*Stmt*.valid ↑ := *Return*.valid ↑; $\big]$

**b1.** *Decl* : *Type Var*
$\big[$ *Decl*.symTabOut ↑ := {*Var*.id : *Type*.name};
*Decl*.attrOut.isUsed[*Var*] ↑ := false;
*Decl*.attrOut.isInitialized[*Var*] ↑ := false;
*Decl*.valid ↑ := true $\big]$

**b2.** *ObjInit* : *Type*$0 *Var* **= new** *Type*$1 *ArgList*
$\big[$ *ArgList*.symTab ↓ := *ObjInit*.symTab ↓;
*ObjInit*.symtabOut ↑ := {*Var*.id : *Type*.name};
*ArgList*.typeList ↓ := *Type*.params ↑;
*ObjInit*.attrOut.isInitialized[*Var*] ↑ := true;
*ObjInit*.attrOut.isUsed[*Var*] ↑ := false;
*ObjInit*.valid ↑ := *ArgList*.valid ↑;
∧ *Type*$0.name ↑ := *Type*$1.name ↑; $\big]$

**b3.** *Invoke* : *Var*$0 = *Var*$1 *Call InvokeMore*
$\big[$ *InvokeMore*.symTab ↓ := *Invoke*.symTab ↓;
*InvokeMore*.exprType ↓ := *Call*.retType ↑;
*Call*.attrIn ↓ := *Invoke*$0.attrIn ↓;
*InvokeMore*.attrIn ↓ := *Call*.attrOut ↑;
*Invoke*.attrOut.isUsed[*Var*$0] ↑ := true;
*Invoke*.attrOut.isUsed[*Var*$1] ↑ := true;
*Invoke*.attrOut ↑ := *InvokeMore*.attrOut ↑;
*Invoke*.valid ↑ := *InvokeMore*.valid ↑
∧ (*InvokeMore*.retType ↑==
   *Invoke*.symTab ↓ [*Var*$0.id ↑])
∧ *Call*.exprType ↑ ==
   *Invoke*.symTab ↓ [*Var*$1.id ↑]; $\big]$

**b4a.** *InvokeMore*$0 : *Call InvokeMore*$1
$\big[$ *InvokeMore*$1.symTab ↓ := *InvokeMore*$0.symTab ↓;
*InvokeMore*$1.exprType ↓ := *Call*.returnType ↑;
*Call*.symTab ↓ := *InvokeMore*$0.symTab ↓;
*InvokeMore*$0.retType ↑ := *InvokeMore*$1.retType ↑;
*Call*.attrIn ↓ := *InvokeMore*$0.attrIn ↓;
*InvokeMore*$1.attrIn ↓ := *Call*.attrOut ↑;
*InvokeMoreOut*$0.attrIn ↑ := *InvokeMoreOut*$1.attrIn ↑;
*InvokeMore*$0.valid ↑ := *Call*.valid ↑
   ∧ *InvokeMore*$1.valid ↑;
   ∧ *Call*.exprType ↑ := *InvokeMore*$1.exprType ↓; $\big]$

**b4b.** *InvokeMore* : ϵ
$\big[$ *InvokeMore*.retType ↑ := *InvokeMore*.exprType ↓;
*InvokeMore*.attrIn.itrVec ↑= (false, false);
*InvokeMore*.valid ↑ := true; $\big]$

**b5** *Call* : *Api ArgList*
$\big[$ *ArgList*.symTab ↓ := *Call*.symTab ↓;
*ArgList*.typeList ↓ := *Api*.params ↑;
*Call*.retType ↑ := *Api*.retType ↑]
*Api*.attrIn ↓ := *Call*.attrIn ↓;
*Call*.attrOut ↑ := *Api*.attrOut ↑;
*Call*.exprType ↑ := *Api*.exprType ↑);
*Call*.valid ↑ := *ArgList*.valid ↑ $\big]$

**b6a.** *ArgList*$0 : *Var ArgList*$1
$\big[$ *ArgList*$1.symTab ↓ := *ArgList*$0.symTab ↓;
*ArgList*$1.typeList ↓ := *ArgList*$0.typeList[1 :] ↓;
*ArgList*$1.attrOut.isUsed[*Var*] ↑ := true;
*ArgList*$0.valid ↑ := *ArgList*$1.valid ↑
   ∧ (*ArgList*$0.symTab ↓ [*Var*.id ↑]
      == *ArgList*$0.typeList[0] ↓); $\big]$

**b6b.** *ArgList* : ϵ
$\big[$ *ArgList*.valid ↑ := *ArgList*.typeList.$isEmpty()$ ↑; $\big]$

**b7.** *Return* : **return** *Var*
$\big[$ *Return*.attrOut.retStmtGenerated ↑ :: true
*Return*.valid ↑ := *Return*.methodRetType ↓==
   *Return*.symTab ↓ [*Var*.id ↑]$\big]$;

**c1.a.** *Stmt* : *Branch*
$\big[$ *Branch*.symTab $\downarrow$ := *Stmt*.symTab $\downarrow$;
*Stmt*.valid $\uparrow$ := *Branch*.valid $\uparrow$;
*Branch*.attrIn $\downarrow$ := *Stmt*.attrIn $\downarrow$;
*Stmt*.attrOut $\uparrow$ := *Branch*.attrOut $\uparrow$; $\big]$

**c1.b.** *Stmt* : *Loop*
$\big[$ *Loop*.symTab $\downarrow$ := *Stmt*.symTab $\downarrow$;
*Stmt*.valid $\uparrow$ := *Loop*.valid $\uparrow$;
*Loop*.attrIn $\downarrow$ := *Stmt*.attrIn $\downarrow$;
*Stmt*.attrOut $\uparrow$ := *Loop*.attrOut $\uparrow$; $\big]$

**c1.c.** *Stmt* : *Except*
$\big[$ *Except*.symTab $\downarrow$ := *Stmt*.symTab $\downarrow$;
*Stmt*.valid $\uparrow$ := *Except*.valid $\uparrow$;
*Except*.attrIn $\downarrow$ := *Stmt*.attrIn $\downarrow$;
*Stmt*.attrOut $\uparrow$ := *Except*.attrOut $\uparrow$; $\big]$

**c2.** *Branch* : **if** *Cond* **then** *Stmt*$1 **else** *Stmt*$2
$\big[$ *Cond*.symTab $\downarrow$ := *Stmt*.symTab $\downarrow$;
*Stmt*$1.symTab $\downarrow$ := *Cond*.symtabOut $\uparrow$;
*Stmt*$2.symTab $\downarrow$ := *Cond*.symtabOut $\uparrow$;
*Branch*.valid $\uparrow$ := *Cond*.valid $\uparrow \wedge$ *Stmt*$1.valid $\uparrow$
$\wedge$ *Stmt*$2.valid $\uparrow$;
*Cond*.attrIn $\downarrow$ := *Branch*.attrIn $\downarrow$;
*Stmt*$1.attrIn $\downarrow$ := *Cond*.attrIn $\downarrow$;
*Stmt*$2.attrIn $\downarrow$ := *Cond*.attrIn $\downarrow$;
*Branch*.attrOut $\uparrow$ := *Branch*$1.attrIn $\downarrow$; $\big]$

**c3.** *Loop* : **while** *Cond* **then** *Stmt*
$\big[$ *Cond*.symTab $\downarrow$ := *Stmt*.symTab $\downarrow$;
*Stmt*.symTab $\downarrow$ := *Cond*.symTabOut $\uparrow$;
*Loop*.valid $\uparrow$ := *Cond*.valid $\uparrow \wedge$ *Stmt*.valid $\uparrow$;
*Cond*.attrIn $\downarrow$ := *Loop*.attrIn $\downarrow$;
*Stmt*.attrIn $\downarrow$ := *Cond*.attrOut $\uparrow$;
*Loop*.attrOut $\uparrow$ := *Loop*.attrIn $\downarrow$; $\big]$

**c4.** *Except* : **try** *Stmt Catch*
$\big[$ *Stmt*.symTab $\downarrow$ := *Except*.symTab $\downarrow$;
*Catch*.symTab $\downarrow$ := *Stmt*.symTabOut $\uparrow$;
*Except*.valid $\uparrow$ := *Stmt*.valid $\uparrow \wedge$ *Catch*.valid $\uparrow$;
*Stmt*.attrIn $\downarrow$ := *Except*.attrIn $\downarrow$;
*Catch*.attrIn $\downarrow$ := *Stmt*.attrIn $\downarrow$;
*Except*.attrOut $\uparrow$ := *Except*.attrIn $\downarrow$; $\big]$

**c5a.** *Catch*$0 : **catch**(*Type*) *Stmt*; *Catch*$1
$\big[$ *Catch*$1.symTab $\downarrow$ := *Catch*$0.symTab $\downarrow$;
*Stmt*.symTab $\downarrow$ := *Catch*$0.symTab $\downarrow$;
*Stmt*.attrIn $\downarrow$ := *Catch*$0.attrIn $\downarrow$;
*Catch*$1.attrIn $\downarrow$ := *Stmt*.attrIn $\downarrow$;
*Catch*$0.attrOut $\uparrow$ := *Catch*$1.attrOut $\uparrow$;
*Catch*$0.valid $\uparrow$ := *Stmt*.valid $\uparrow \wedge$ *Catch*$1.valid $\uparrow$; $\big]$

**c5b.** *Catch* : $\epsilon$
$\big[$ *Catch*.valid $\uparrow$ := true; *Catch*.attrOut $\uparrow$ := $\phi$ $\big]$

**c6.** *Cond* : *Call*
$\big[$ *Call*.symTab $\downarrow$ := *Cond*.symTab $\downarrow$;
*Cond*.valid $\uparrow$ := *Call*.valid $\uparrow$;
*Call*.attrIn $\downarrow$ := *Cond*.attrIn $\downarrow$;
*Cond*.attrOut $\uparrow$ := *Call*.attrOut $\uparrow$; $\big]$

**d1.** *Api* : **JAVA_API_CALL**
$\big[$ *Api*.name $\uparrow$ := $NAME$;
*Api*.params $\uparrow$ := $FORMAL\_PARAM\_LIST$;
*Api*.exprType $\uparrow$ := $TYPE$;
*Api*.retType $\uparrow$ := $RET\_TYPE$;
if(*Api*.name == "hasNext")
   *Api*.attrOut.itrVec $\uparrow$ := $(\text{true}, \text{false})$;
else if(*Api*.name == "next")
   *Api*.attrOut.itrVec[1] $\uparrow$ := true; $\big]$

**d2.** *Api* : **INTERNAL_METHOD_CALL**
$\big[$ *Api*.name $\uparrow$ := $NAME$;
*Api*.params $\uparrow$ := $FORMAL\_PARAM\_LIST$;
*Api*.exprType $\uparrow$ := $\epsilon$;
*Api*.retType $\uparrow$ := $RET\_TYPE$;
*Api*.attrOut.itrVec $\uparrow$ := *Api*.attrIn.itrVec $\downarrow$; $\big]$

**d3.** *Type* : **JAVA_TYPE**
$\big[$ *Type*.name $\uparrow$ := $NAME$
*Type*.params $\uparrow$ := $FORMAL\_PARAM\_LIST$; $\big]$

**d4.** *Var* : **VAR_ID**
$\big[$ *Var*.id $\uparrow$ := $ID\_NUMBER\big]$