# OpenReview forum: "Neural Program Generation Modulo Static Analysis"
_NeurIPS.cc/2021/Conference — NeurIPS 2021 Spotlight_

### Official Review · Reviewer_spNu · 2021-07-14

**Rating:** 9
**Confidence:** 4

**Summary:**

Leveraging compiler inferences about code, such as from static analyzers, can improve the ability of neural models to reason about code. This paper introduces a method that relies on attribute grammar coupled with static analysis that operates on partial programs, demonstrating that access to this information greatly improves the quality of generated programs.

**Limitations And Societal Impact:**

The authors have adequately addresses both.

**Main Review:**

*Note: I appreciate the authors' response; they did not impact my score because I already strongly support acceptance.*

This is a strong paper. It is well-written and easy to follow while introducing a method that bridges several ideas and performs well. The resulting modeling delta compared to previous models is simple enough that it ought to be more broadly useful, although, as the authors point out as well, the need to implement static analyzers is a bit of a hurdle. Given that, my review mostly concerns minor issues and notes.

In terms of evaluation, while the baselines seem adequate, it is a slightly questionable choice to apply CodeGPT in this way -- presenting entire files while removing the method's body. Besides being somewhat removed from its pretraining objective (which the finetuning presumably helped overcome), transformers typically don't scale well to larger inputs. Indeed, the appendix suggests that this model was trained with a sequence length of up to 512 tokens, which tends to be (far) less than the size of typical files. Please clarify how this impacted the model.

Presentation:
- L147: consider splitting out the definition of mu and sigma from this equation, to make it more readable.
- The citation style used in the paper appears to be incorrect for NeurIPS; it uses the numerical references rather than surname-of-first-author.
- Table 1: consider adding a "totals" row, to make it easier to assess the overall performance of these models. Also consider better emphasizing the "intermediate" totals (row 5 & 10).

Typos:
- L47: "and whose with grammar"
- L390: missing semicolon or "and" after "beam search"


**Time Spent Reviewing:**

1.5

---

> ### Author Response · Authors · 2021-08-10
> **Author Response to Reviewer spNu**
>
> Thank you for the review. We address the main points below:
>
> ```
> > CodeGPT was not designed to be used in this way.
> ```
>
>
> It is true that CodeGPT was not designed for this kind of whole-method-body generation. However, we needed a transformer baseline to compare with, and CodeGPT was the most reasonable transformer baseline we found. We tried our best to make it fit our setup. That said, after reading the reveiwer's comment, we are sensitive to the possibility our experiments portray CodeGPT in a negative light, when we are in fact using it in a way that its designers did not consider. When revising the paper, we will make the imperfect match between CodeGPT and our task clearer, to ensure we do not unfairly impugn the method.
>
>
> ```
> > The appendix suggests that this model was trained with a sequence length of up to 512 tokens, which tends to (far) exceed the size of typical files. Please clarify how this impacted the model.
> ```
>
> We chose a window size of 512 as this number is common in transformer models of code (for example, the CodeBERT paper uses a window of 512 as well). As mentioned in our response to R3 (3UN5), candidate training examples that required a window longer than 512 tokens were eliminated. This criterion reduced the size of our training set by 30%, but still left us with a training dataset of size 3.3M.
>
>
> ```
> > Table 1: consider adding a "totals" row, to make it easier to assess the overall performance of these models. Also consider better emphasizing the "intermediate" totals (row 5 & 10).
> ```
> This is a great suggestion, and we will follow it while preparing the revision.

---

### Official Review · Reviewer_3UN5 · 2021-07-15

**Rating:** 7
**Confidence:** 3

**Summary:**

This paper presents a new model for automatic code completion of full methods
(rather than simply single lines).  The main innovation is the use of a static
analysis tool which can analyze partially generated programs.  The program is
generated hierarchically from an attribute grammar using a model they call a
Neurosymbolic Attribute Grammer (NSG).  At each step the model has access to:
- The context code surrounding the code to be generated
- All the tree nodes which have been generated so far
- The attributes generated by running the static analyzer over the partially
completed program

Their hypothesis is that the kind of information that the static analyzer
generates is difficult for a neural network model to learn by simply training on
a large dataset of programs.  They show that their model outperforms both a
smallish transformer baseline (~100M parameters) and a Graph Neural Network
baseline.


**Ethical Concerns:**

I don't see any ethical concerns which are not shared by all code generation models trained on publicly available data.

**Limitations And Societal Impact:**

I would have liked to see the author acknowledge the fact that their baseline model was a relatively small transformer and that larger transformers may significantly reduce/eliminate the gains provided by their methods.

**Main Review:**

Strengths:
- The paper is well written and relatively easy to understand.
- Their model *significantly* outperforms all of the baseline models on both
their static analysis metrics as well as their whole code metrics.
Specifically, the transformer baseline CodeGPT only generates correct API calls
for 4.7% of test samples while their method generates correct API calls for 43.7% of test samples.
- Tackles the important and immediately applicable task of conditional code
generation.
- The idea of using on-line static analysis to aid neural models seems very
generally applicable to almost any conditional code generation model, and their
results show that it can make a significant difference.


Weaknesses:
- They only test on relatively small transformer models.  Recent work has
shown that scaling the size of such transformer models significantly improves
their performance.  So it's possible that the gains from the static
analysis will disappear once the transformer models are big enough.  While I don't expect them to scale to GPT-3 results it would have been good to see some analysis of how the performance gains change as the Transformer model size increases.
- As far as I can tell their model is trained on canonicalized versions of
the code where all variable, method and class names are changed to generic names
(i.e. var_0, var_1, etc.), which prevents the model from taking advantage of the
soft information provided in such naming in source code corpora.
- Most of their whole method metrics are AST based metrics which are biased
toward AST base models like theirs rather than sequential metrics like BLEU
which would be biased towards sequential models like CodeGPT (the transformer
baseline).  I understand that code is inherently hierarchical and the AST
metrics may better represent this, but I would have liked to see results for
BLEU score as well (which should be quite easy to compute).
- Their evaluation setup does not include test cases so they cannot directly
execute the code to tell whether the code they generate is semantically correct
(or at least semantically matches the code in the dataset).

Questions:
- Was the context size of 512 tokens that you used for CodeGPT large enough
to include both the relevant specification (surrounding code) as well as the
generated method body for most samples in your training and test sets? What
fraction of samples were truncated in the training and test sets?
- What were you trying to show with the 50% evidence visible setting?  This
doesn't seem like a realistic setting and so I wasn't sure why you included
these results.

**Time Spent Reviewing:**

4

---

> ### Author Response · Authors · 2021-08-10
> **Author Response to Reviewer 3UN5**
>
> Thank you for your comments. We address the main points in the review below:
>
> ```
> >  It is possible that the gains from the static analysis will disappear once the transformer models are big enough.
> ```
>
> This may indeed be the case. That said, the transformer baseline tested (CodeGPT) already has an advantage over our model in terms of size. Specifically, our model has ~63M parameters, whereas CodeGPT has ~124M parameters. While the performance of the transformer baseline is likely to go up with more parameters, we could also provide commensurately more parameters to our learning model and make it perform better.
>
> The question, then, is: would the growth in the performance of the transformer with increased size far outpace our model? This is possible, and should be investigated in future work.
>
> As a separate point, we believe that parameter-efficiency in a model can be valuable in itself. For example, one could train such a model on a single GPU (as we have done in this work) and run it on a memory-limited device. The prior knowledge needed to achieve this  parameter-efficiency comes for free (from the language semantics) in our setting.
>
> ```
> > We change local variable and method names to generic names such as var_1. This prevents our model from learning the soft information provided by variable names.
> ```
>
>
> This is a very good point.  On one hand, there is (likely) an advantage to treating variable names as tokens that are part of a grammar: it makes it easy to include information about variables in the attribute grammar, and presumably easier for the model to learn to use them.  That said, this comes at a cost: "var_1" has less information about programmer intent than "myFileReader".  Retaining this soft information somehow may boost accuracy, and is a good idea to consider.
>
> ```
> > Why not use BLEU scores as the evaluation metric?
> ```
> There are some basic problems with using BLEU scores in the code generation setting. We use one of the examples in Table 3 of the paper ("reading from a file") to illustrate this point. Here, the ground-truth code is:
>
> ```
> public String reader(){
> 	StringBuffer stringBuffer = new StringBuffer();
> 	String line;
> 	while ((line = bufferedReader.readLine()) != null) {
> 		stringBuffer.append(line);
> 		stringBuffer.append("\n");
> 	}
> 	return stringBuffer.toString();
> }
> ```
>
> The CodeGPT output is:
>
> ```
> public String reader(){
> 	StringBuffer buffer=new StringBuffer();
> 	buffer.append("\n");
> 	return buffer.toString();
> }
> ```
> The NSG output is:
>
> ```
> public String reader(){
> 	java.lang.String var_9;
> 	try {var_9=field_5.readLine();
> 	} catch(IOException var_8) {
> 		var_8.printStackTrace();
> 	}
> 	return var_9;
> }
> ```
>
> The NSG output is clearly better. However, the CodeGPT output gets a higher BLEU score because it uses variable names that superficially match the ground-truth.  Specifically, the BLEU score of the CodeGPT output is 25.11 and the BLEU score of the NSG output is 19.07.
>
>
> This kind of situation arose often in our tasks, and this is why we did not use BLEU score as a metric.
>
> ```
> > The evaluation does not include test cases on which the code is directly executed.
> ```
> This is a great point. The static analysis checks performed during our evaluation are an approximate semantics check. But this checking could be expanded with stronger specifications such as test cases. We are actively exploring this direction in current work using a combination of classical program synthesis and NSG-style models.
>
> ```
> Was the context size of 512 tokens, used for CodeGPT, large enough to include both the relevant specification (surrounding code) as well as the generated method body?
> ```
>
>
> The CodeGPT model was trained on a sequence of tokens consisting of class name, class-instance variables, and sequence of method headers of surrounding code, followed by the method header of the unknown method. The training is on the method header (and the original method body does not count toward the 512 tokens). Candidate training examples (without the method body) that were longer than 512 tokens were eliminated from the training set. This criterion for selecting training examples reduced the size of our training set by 30%. However, we had a dataset of size 3.3M training examples even after this reduction, which we believe is sufficient for fine tuning.

---

> > ### Comment · Reviewer_3UN5 · 2021-08-20
> > **Thanks for your response.**
> >
> > I wanted to thank the authors for their response.
> >
> > While I agree that parameter efficiency in itself can be useful I think it's important to explore whether that's the main advantage of a technique or whether we also expect it to work better at scale.  As large models provide state-of-the-art results in more and more settings, I think it's important for papers to attempt to project how their performance  gains would change with increased scale, even if they don't have the resources to actually test at very large scale.
> >
> > None-the-less I still think this is a good paper and should be accepted.

---

> > > ### Author Response · Authors · 2021-08-25
> > > **Thank you!**
> > >
> > > We agree that there is value in understanding the relative benefit of our approach as the transformer baseline grows in size. Over the last week, we have been exploring concrete ways to evaluate this. Specifically, we have begun working on an experiment in which we increase the number of parameters of the CodeGPT baseline by a factor of k (k <= 10, given our computational budget). These larger models use the same pretrained layers (roberta-base) as CodeGPT but have more subsequent layers, and are finetuned on the same Context -> Java method dataset that we use to finetune CodeGPT. In the final version of the paper, we can report on the gains in the transformer's performance as the model size is increased this way. While this experiment will not answer your question conclusively, we hope that it will provide some further clarity to the readers of the paper.

---

> > > > ### Comment · Reviewer_HvAo · 2021-08-25
> > > > **Try a different Transformer**
> > > >
> > > > *[Note: different reviewer.]*
> > > >
> > > > As an aside, I'd recommend replacing CodeGPT in this experiment with a different Transformer – for instance, with the two smallest checkpoints of GPT-NEO, which should fit your budget, or with GPT-J if you're willing to adopt a bit of engineering. This is for several technical but practically important reasons:
> > > >
> > > > - GPT-NEO is pretrained on a large dataset of language+code, CodeGPT on a smaller one just of code. Pretraining on language seems to matter substantially for code completion, in part thanks to plethora of code naming/documentation conventions we follow in the context.
> > > > - GPT-NEO follows the common subword tokenization strategy, which vastly simplifies joint modeling of language and code.
> > > > - In fact, the preprocessing pipeline of CodeGPT removes quite a bit of code formatting and structure that appears to matter in the long run. It matters less for Java than for something like Python, but still makes a difference. Fine-tuning on properly formatted data simply makes the network forget everything it learned in pretraining, and as such, simply does not leverage the pretrained checkpoint. That does not matter on smaller sizes (in effect, you're just training a Transformer from scratch), but after some threshold of model size there's a meaningful difference in convergence speed between starting from a pretrained checkpoint or from scratch.
> > > > - Various sharding, half-precision, and offloading techniques in [DeepSpeed](https://huggingface.co/transformers/master/main_classes/deepspeed.html) allow one to scale up inference and tuning substantially even on a single GPU. You should be able to tune a very strong baseline Transformer of size ~12B. If you adopt these techniques, GPT-J is a much, much better baseline than GPT-NEO.
> > > >
> > > > While these might seem like secondary engineering decisions, *this* paper is a great opportunity to establish a meaningful comparison between purely-neural and neuro-symbolic approaches to code generation. In order for that comparison to be trusted and appreciated by the community, its neural baseline would ideally be as strong and large as possible.

---

> > > > > ### Author Response · Authors · 2021-08-26
> > > > > **Yes, agreed**
> > > > >
> > > > > Yes, we agree. As mentioned in our response to your (HvAo) main review, we will do additional comparisons with GPT-Neo in the final version. We have also begun looking into GPT-J and will make a good-faith attempt to compare against it.

---

### Official Review · Reviewer_bycS · 2021-07-16

**Rating:** 6
**Confidence:** 4

**Summary:**

The paper proposes a novel approach to conditional program generation that leverages supervision from a static analysis tool during generation. This serves to address a common difficulty in program synthesis using language models, that of generating code without semantic errors. The program generation and static analysis both follow the linear information flow through a program defined by an attribute grammar, with the program generation model having access to the attributes from the analysis. The paper applies this technique to a dataset of 1 million Java programs, comparing primarily with baselines CodeGPT and GNN2NAG. Evaluating both on static checks, overall fidelity metrics, and exact match, the paper finds the proposed NSG method outperforms the baselines considered on nearly all metrics.

**Limitations And Societal Impact:**

The authors discuss the main technical limitation of the proposed approach, that the implementation is language-specific, whereas language model baselines are generally language agnostic. The paper does not address (nor, per the checklist, anticipate) potential negative societal impact.

**Main Review:**

The paper considers the challenging and important task of conditional program generation, an area of increasing interest at NeurIPS in recent years. The main weakness identified, that neural models of source code usually struggle with generation of long sections of code, is important. Being able to generate long semantically coherent programs is a key challenge in the field, and this paper presents an approach to program generation that deals with this effectively.

The method is explained clearly overall, providing background on static analysis with attribute grammars and a clear explanation of the algorithm for using such a static analysis during program generation, compatible with a wide class of models. The appendix provides the details that were omitted from the main paper, including how the training data is prepared, the attributes considered and part of the definition of the attribute grammar, how attributes are encoded, and details about the main and baseline methods.

A key element of the approach is the attribute grammar defines a linear path through the program, allowing the model to be trained using pre-computed attributes and requiring on-the-fly computation of attributes during inference. This is explained best in the training and inference appendix.

A strength of the approach, and an advantage over prior work, is that the method allows for learning constraints and guiding principles from the input corpus, rather than using the static analysis to provide hard constraints. This gives more freedom to the static analysis designer to include attributes that correspond to best-practices or that appear only in certain contexts, without needing to avoid attributes that don't define hard-and-fast rules about programs. Prior work leveraging attribute grammars for program generation used them to enforce hard constraints.

A weakness is that the evaluation static checks match those properties provided by the static analysis attribute grammar. I would be interested to see an evaluation indicating whether the method learns to preserve additional properties beyond those directly measured by the static analysis. Perhaps to that end, please clarify what is meant by 50% evidence. Does this refer to full evidence at 50% of nodes, or 50% evidence at each node? How is the 50% selected? Outside of these concerns, the evaluation is solid. The task and baselines are appropriate, the metrics capture properties of interest, and the evidence from the results tells a quite clear story. Including a measure of spread would improve the results section.

---

Line 71: $\delta$ accepts programs as inputs, but in the prediction error expression $\textbf{E}_{(X,Y)\sim D}[\delta(P_\theta(X|Y), Y)]$ you incorrectly pass $P_\theta(X|Y)$, rather than a program, as the first input to $\delta$.

Nit: Line 165 refers to specification Y rather than specification X.

Line 765 has a broken reference simply appearing as "??".

**Time Spent Reviewing:**

8

---

> ### Author Response · Authors · 2021-08-10
> **Author Response to Reviewer bycS**
>
> We really appreciate your feedback. We address the main points in the review below:
>
> ```
> > Can our model preserve additional properties beyond those directly measured by the static analysis?
> ```
>
> As the reviewer implies, there may be families of program properties that are closely related to each other. For a simple example, type-safe Java programs do not read from uninitialized variables. This means that a model trained on type-safety attributes will be likely to produce programs in which all variables are initialized before use.
>
> One way to evaluate this question will be through an ablation study in which we reduce the set of attributes available to the model as inputs during training and inference, but we check all the properties that we currently check, to see if some of the properties are satisfied "for free". If the reviewer feels this is useful, we can perform such a study for the final version of the paper.
>
> ```
> > What does "50% evidence" mean?
> ```
>
> First, let us re-state that "evidence" (as defined in Section 2) refers to the context in the surrounding code (method names, method headers, Java Doc comments, class variables, etc.) upon which generation of a Java method is conditioned. Suppose we are trying to generate a method body M. When we use "x% evidence" for this task, each piece of evidence in the code surrounding M is selected with independent probability x% to be part of the evidence set.
>
> For example, if a test class has 4 methods, their names can each be used as evidence from the surrounding context to predict the body of the 5th method M. In the "50% evidence" scenario in our experiments,  each of these 4 names has a 50% chance of being part of the "method names" evidence set. The use of such incomplete evidence sets is inspired by real-world programming scenarios in which a user wants to auto-complete the next method in a class in which much of the code is unwritten.
>
> We agree that these points were not fully clear in the draft we submitted. We will clarify this in more detail in the final version with clear examples.
>
>
> ```
> > Typos/errors in Line 71, 165 and 765
> ```
> Thank you for pointing out these errors. We will correct them in the revision.

---

> > ### Comment · Reviewer_bycS · 2021-08-24
> > **Thanks for your response**
> >
> > Thank you for your response. I appreciate the clarification of the meaning of "X% evidence", and I agree that incomplete evidence sets are more representative of the the real-world scenario of interest than complete evidence sets.

---

### Official Review · Reviewer_HvAo · 2021-07-19

**Rating:** 8
**Confidence:** 5

**Summary:**

The system generates long code snippets such as method bodies given their surrounding context. In contrast to modern
code generation models that are supervised by program syntax, it additionally conditions generation on semantic
attributes that are computed on AST nodes by the compiler. The paper shows that this approach generate significantly
more idiomatic code with fewer language errors than syntax-only Transformers or purely-neural neural attribute models.

**Limitations And Societal Impact:**

Limitations are explicitly and nicely discussed in the Conclusion.

On societal impact, the authors state "We have proposed an approach to better generation of source code and used open-source programs as training data. We do not anticipate any direct negative societal impact of this work."
As a counter-point, I recommend reading the thorough and elaborate discussion of possible impacts of such models in the very recent "Evaluating Large Language Models Trained on Code" (Chen et al., 2021).

**Main Review:**


Strengths:

+ A principled formalism and implementation that incorporates semantic information computed by the compiler into neural
  code generation models. Prior work usually either ignores it (syntactic codegen, either for sequences or for ASTs),
  hard-filters by it at inference time, or incorporates specific instances, like data flow in GNN-based models. Adopting
  it into the well-known formalism of attribute grammars is appreciated.
+ Clear motivating examples and well-explained decision rationale
+ Fair comparison to two most relevant baselines, one syntax-only and one attribute-inspired but not explicitly
  supervised with symbolic attributes.
+ Evaluation along multiple important dimensions: method generation accuracy, small expression generation accuracy (as
  in prior work), and individual language rules.
+ An explicit discussion of limitations is welcome, not discussed often enough in our community.

Weaknesses:

- The paper spends significant space on the NSG formalism and on the particular generative model re-used from Murali et
  al., which forces it to defer its instantiation for Java, choice of attributes, and their encoding to the Appendix.
  As a result, Section 4 ends abruptly, confusing the reader. The "design" paragraph helps set up the scene but, I
  imagine, would not actually be informative to someone without formal methods background, at least without a single
  concrete example.
- Similarly, lack of space likely forced the authors to leave out Appendix H. It provides a valuable complementary
  perspective on the respective strengths and weaknesses of training approaches, and a quick summary of it would be
  welcome in the paper.
- Characterizing [7] as "next token prediction" is unfair, as that model generates whole ASTs and – just like this
  work – supervised with its production rule decisions at every step. The difference is in (a) size of the target AST,
  (b) the amount of available in-method context for conditioning and for copy.
- Phrasing aside, it would be interesting to see how [7] fairs against NSG for _in-method_ expression completion (medium
  horizon). I imagine NSG would still outperform, as it leverages more explicit conditioning on attributes rather than
  their neural abstractions.

Questions & Remarks:

- The general line of reasoning that "semantics are hard to learn from syntax" is not wrong, but:
  - It largely depends on the size of the model and its pretraining data, with improvement following a power law.
  - It further improves with multi-language training as the model pick ups non-formalizable yet similarly expressed
    cross-language cues.

  Both were shown empirically by [Codex](https://arxiv.org/abs/2107.03374), concurrently with this submission. This does
  not diminish the paper's contribution, just shows that the two directions keep improving and could further benefit
  each other.

- A better Transformer baseline would be [GPT-NEO](https://github.com/EleutherAI/gpt-neo) or
  [GPT-J](https://github.com/kingoflolz/mesh-transformer-jax/#gpt-j-6b). Both are pretrained on The Pile corpus, which
  includes a big portion of Github and more natural language. The former was shown to be a strong baseline for code
  generation against Codex and in the [APPS paper](https://arxiv.org/abs/2105.09938), with or without finetuning.

- L404-405: There are many methods to integrate tree structure and generative Transformers. See e.g.:
  - Scholak et al. "DuoRAT: Towards Simpler Text-to-SQL Models", NAACL 2021.
  - Sun et al. "TreeGen: A Tree-Based Transformer Architecture for Code Generation", AAAI 2020.

- How much of attribute vocabulary could be shared cross-language? Most popular imperative languages could implement a
  compiler frontend that computes a version of the attributes in Appendix D. Could this help us effectively train
  cross-language and cross the gap to language model training?

- Is the NSG model robust to presence or absence of attributes? For example, suppose the code/linker is in a broken
  state so that type-based attributes cannot be properly resolved. Does the model break down completely or can it still
  rely on other features?

- Follow-up to previous question: In addition to conditioning on attributes, the model could be trained to predict them
  as an auxiliary task. Multitask learning has improved performance or accelerated training in many domains, code could
  work just as well.

Typo:
- L47: "whose with grammar"

**Time Spent Reviewing:**

4

---

> ### Author Response · Authors · 2021-08-10
> **Author Response to Reviewer HvAo**
>
> Thank you for your comments. We respond to the main points in the review below:
>
>
> ```
> > "Characterizing [GNN2NAG] as "next token prediction" is unfair...
> ```
> Thank you for this feedback. We will revise the referenced statement along the lines you suggest.
>
>
> ```
> > It would be interesting to see how GNN2NAG fares against NSG for in-method expression completion (medium horizon).
> ```
> Agreed. We are currently working to set up this experiment and will add the results from it to the final version of the paper.
>
>
> ```
> > We should compare with the GPT-NEO or GPT-J transformers.
> ```
> Thank you for these pointers. We have begun looking into these baselines and will try to experimentally compare against GPT-NEO in the final version.
>
>
> ```
> > Is the NSG model robust to presence or absence of attributes at test time, for example, because the code/linker is in a broken state?
> ```
>
> This is a great question. Our guess is that the NSG model will not work very well if attributes are missing at test time. For example, suppose the model has learned a type safety feature that relates the type of a variable in the symbol table with the way it is used in a production rule. If the variable is missing from the symbol table during testing (because of a broken compiler), the model might take it as a signal to not invoke that rule.
>
> That said, this hunch should be experimentally validated. We will perform this experiment and discuss it in the final paper.
>
>
> ```
> > Section 4 ends abruptly and doesn't discuss the instantiation of NSGs for Java.
> ```
>
>
> We will rewrite Section 4, reducing the overlap with Murali et al. and discussing in more detail how the model is instantiated for Java and specific static analyses.
>
> ```
> > In addition to conditioning on attributes, the model could be trained to predict them as an auxiliary task. Multitask learning has improved performance or accelerated training in many domains, code could work just as well.
> ```
>
> This is a very interesting future direction. It is conceivable that any language model, if applied to code, could see a significant accuracy boost if trained to predict semantic attributes. Semantic attributes could be seen as hidden variables, present during training but absent during inference, where the actual code text is conditioned upon the value of those variables.  There may be advantages to this approach compared to what we have proposed; in partciular, very heavy-weight semantic analyses could be used, that are not appropriate for use at inference time.
>
> ```
> > There are many other methods to integrate tree structure and generative transformers that we should discuss.
> ```
> We agree and will discuss these methods in the related work in the final version.
>
> ```
> > How much of the attribute vocabulary could be shared cross-language?
> ```
> Another interesting question. As it happens, we are actively pursuing cross-language training, built on top of an LLVM-based, source-language-agnostic static analysis infrastructure, as a follow-up research direction. We will mention this ongoing work in the conclusion of the revised paper.
>
> ```
> > The general line of reasoning that "semantics are hard to learn from syntax" is not wrong, but comes with caveats.
> ```
>
>
> We agree. We will discuss these points in more depth, and cite the [Codex](https://arxiv.org/abs/2107.03374) paper,  in the revision.

---

### Decision · Program_Chairs · 2021-09-27

**Decision:**

Accept (Spotlight)

**Comment:**

The paper is well written and clearly motivated, with good results outperforming the most relevant baselines.  All reviewers agree this is a good paper and should be accepted.

There are, however, some comments shared by the reviewers that the authors should take into account to improve this work.  Most notably the transformer language model baseline chosen for comparison may have been a bit weak given the recent progress in this direction.